# Qualitative rather than quantitative phosphoregulation shapes the end of meiosis I in budding yeast

Dunja Celebic[1,2], Irem Polat[1], Véronique Legros [1], Guillaume Chevreux[1], Katja Wassmann [1,2] & Sandra A Touati [1,2]✉

## Abstract

**Exit from mitosis is brought about by dramatic changes in the phosphoproteome landscape. A drop in Cyclin-dependent kinase (Cdk) activity, the master regulatory kinase, and activation of counteracting phosphatases such as Cdc14 in budding yeast, results in ordered substrate dephosphorylation, allowing entry into a new cell cycle and replication licensing. In meiosis however, two cell divisions have to be executed without intermediate DNA replication, implying that global phosphorylation and dephosphorylation have to be adapted to the challenges of meiosis. Using a global time-resolved phosphoproteomics approach in budding yeast, we compared the phosphoproteome landscape between mitotic exit and the transition from meiosis I to meiosis II. We found that unlike exit from mitosis, Cdk phosphomotifs remain mostly stably phosphorylated at the end of meiosis I, whereas a majority of Cdk-unrelated motifs are reset by dephosphorylation. However, inducing an artificial drop of Cdk at metaphase of meiosis I leads to ordered substrate dephosphorylation, comparable to mitosis, indicating that phosphoregulation of substrates at the end of meiosis I is thus mainly qualitatively rather than quantitatively ordered.**

**Keywords** Cdk; Cell Cycle; Meiosis; Metaphase I to Anaphase I Transition
**Subject Categories** Cell Cycle; Post-translational Modifications & Proteolysis; Proteomics

## Introduction

The mitotic cell cycle ensures the equal partitioning of the genome into two daughter cells, by ordering DNA replication in S-phase and cell division in M-phase, thus maintaining integrity of the diploid genome. In meiosis, the order of key cell cycle events has to be adapted to generate four haploid cells—gametes or spores. After a pre-meiotic S-phase, cells undergo two successive divisions, meiosis I (MI) and meiosis II (MII), without an intermediate S-phase. Homologous chromosomes are segregated in MI and sister chromatids in MII. A crucial challenge constitutes the MI-MII transition, when cells have to prevent a second round of DNA replication without interfering with the two rounds of chromosome segregation (Marston and Amon, 2004; Morgan, 2007).

Phosphorylation is one of the most prominent post-translational modifications regulating cell cycle progression. Phosphorylation changes protein interactions, regulates enzymatic activities as well as protein localization. The phosphorylation status of a given substrate is the result of a tight balance between kinases and their opposing phosphatases, whose activities must be precisely controlled to shape the entire phosphoproteome of the cell. Among the kinases regulating cell cycle events, Cyclin-dependent kinases (Cdks) are the master cell cycle kinases. To be active, Cdks needs to bind to a regulatory subunit called cyclin, and whose identity depends on the cell cycle stage. Whereas higher eukaryotes harbor several, cell-stage specific Cdks and cyclins, budding yeast harbors only one Cdk named Cdc28, and nine mitotic cyclins (Cln1-3 and Clb1-6). Successive waves of cyclin expression and degradation regulate the mitotic cell cycle, but the expression patterns of cyclins are more complex in meiosis. Clb3 and Clb4 expression overlap in mitosis, while in meiosis Clb4 is active in meiosis I and II and Clb3 expression is restricted to meiosis II. Clb2, which gives rise to Cdk harboring maximal activity in mitosis, is transcriptionally repressed in meiosis and substituted by its paralog Clb1, which is foremost active in meiosis I (Carlile and Amon, 2008; MacKenzie and Lacefield, 2020; Morgan, 2007). In addition to phosphorylation events mediated by Cdks, ubiquitin-dependent degradation plays a key role in regulating protein levels of key factors. For chromosome segregation to take place, the E3 ubiquitin ligase Anaphase Promoting Complex/Cyclosome (APC/C) targets key cell cycle regulators such as cyclins, for degradation (Bloom and Cross, 2007).

In budding and fission yeast, two models have been proposed to ensure timely activation and inhibition of sequential biological events such as DNA replication in S-phase and chromosome segregation in M-phase. The qualitative model relies on cyclin substrate specificity to activate or inhibit specific substrates at the right time. The quantitative model is based on distinct Cdk

[1]Université Paris Cité, CNRS, Institut Jacques Monod, 75013 Paris, France. [2]Sorbonne Université, CNRS, Institut de Biologie Paris Seine, IBPS, UMR7622 Paris, France. ✉E-mail: sandra.touati@ijm.fr

thresholds that control progression through the different cell cycle phases (Coudreuse and Nurse, 2011; Stern and Nurse, 1996). Cdk activity is low in G1/S and increases to reach the final threshold required for M-phase entry and concomitant S-phase inhibition. In anaphase, Cdk activity decreases to allow mitotic exit. To complete this model, the ratio between Cdk activity and its main opposing phosphatase, Cdc14, allows for sequential substrate dephosphorylation to reset the cell cycle at mitotic exit in budding yeast (Bouchoux and Uhlmann, 2011; Kuilman et al, 2015; Powers and Hall, 2017; Touati et al, 2019).

Kinases and phosphatases recognize their targets using a code that relies on several features, including the residues (generally a serine or a threonine), the phosphomotifs surrounding the phosphosites, the docking regions located at a distance from the phosphosite, as well as their accessibility depending on their location in ordered or disordered substrate regions (Holt et al, 2009; Ord et al, 2019; Valverde et al, 2023). The code determines if and when the phosphorylation status of a substrate changes as well as the dynamics of the phosphorylation event. The Cdk-cyclin complex phosphorylates serine and threonine residues on full (S/T) Px(K/R) and minimal (S/T)P consensus motifs (Holt et al, 2009; Suzuki et al, 2015; Ubersax et al, 2003). Two other essential mitotic kinases, Cdc5 (Plk in mammals) and Ipl1 (Aurora in mammals), prefer the phosphomotifs (D/E/N)x(S/T) and (K/R)(K/R)x(S/T), respectively (Alexander et al, 2011; Mok et al, 2010). Like kinases, also phosphatases are multimeric enzymes that recognize specific consensus sites. Their catalytic pocket preference changes upon their binding to distinct regulatory subunits. For example, the phosphatase PP2A$^{Cdc55}$ (PP2A-B55 in vertebrates) preferentially dephosphorylates threonine over serine, thereby impacting substrate dephosphorylation kinetics (Bremmer et al, 2012; Hoermann et al, 2020; Kokot and Kohn, 2022; Kruse et al, 2020; Touati et al, 2019).

In meiosis, one M-phase is followed by the next. After the first meiotic division and in contrast to mitotic exit, the cell cycle must not be fully reset. We call this particular cell cycle transition "meiosis I exit". At meiosis I exit, licensing of DNA replication and chromosome segregation are uncoupled (Phizicky et al, 2018). This raises important questions about how the global phosphoproteome landscape at the MI-MII transition differs from mitotic exit. In particular, it is currently unknown whether the MI-MII transition depends on cyclin specificity or on the Cdk threshold.

Here, we set out to compare exit from mitosis with exit from meiosis I in budding yeast. Previously, we have shown that sequential waves of kinase consensus site dephosphorylation order cell cycle events during mitotic exit (Touati et al, 2018). In the present study, using a global and time-resolved phosphoproteomic approach in meiosis, we compare the phosphoproteome landscape between mitotic and meiosis I exit in a wild type context. Unexpectedly, our study reveals opposite patterns of kinase consensus motif phosphorylation when comparing mitotic and meiosis I exit. The phosphorylation of full Cdk motifs does not drop at the MI-MII transition and we do not observe temporal ordering of phosphomotif dephosphorylation, such as in mitosis. Crucially, mimicking mitotic exit by inducing complete Cdk decrease in meiosis I recapitulates the phosphorylation patterns and the temporal order of phosphomotif dephosphorylation characteristic of mitotic exit. This indicates that at the MI-MII

transition substrate dephosphorylation is mainly ordered qualitatively rather than quantitatively in budding yeast.

# Results

## The phosphorylation landscape of the MI-MII transition in budding yeast

To study the phosphorylation dynamics of meiosis I exit in a time-resolved manner, we took advantage of the GALpromoter-NDT80 block-release system (Carlile and Amon, 2008). Ndt80 is a meiosis-specific transcription factor, which activates the transcription of the middle class meiotic genes. In this system, the open reading frame of Ndt80 is placed under the control of the inducible GAL1-10 promoter. The concomitant expression of a Gal4 estrogen receptor fusion protein (Gal4.ER) induces transcription from the GAL1-10 promoter after the addition of estrogen to the culture media. In the absence of estrogen (β-estradiol), Ndt80 is transcriptionally repressed, and cells stay arrested in pachytene stage of prophase I. Upon addition of β-estradiol, middle class meiotic genes are transcribed and cells are released from pachytene with excellent synchrony. FACS (fluorescence-activated cell sorting) was used to monitor the synchrony of the pre-meiotic cell cycle from G0 to G2 stages, and visualization of DNA and spindles by immunofluorescence to assess progression from prophase I to anaphase II (Fig. 1A). Cells went through S-phase synchronously 2 h after washing them in sporulation media and the genome was duplicated after 3 h. We also monitored by Western Blot the expected increase of Cdk activity from G0 to M-phase by using an antibody recognizing SPx(K/R) phosphomotifs. We observed new bands appearing 2–3 h after sporulation medium release, with a peak reached at 5 h (Fig. EV1A). At 5h30, β-estradiol was added and time points were collected at 10 min intervals to confirm the synchrony of meiotic progression. β-estradiol addition corresponds to time 0 when cells are in prophase I. 50 min later the majority of cells were in metaphase I, at 80 min they entered anaphase I and at 140 min the large majority had reached metaphase II. To quantitatively capture phosphorylation changes during meiotic progression, we employed Tandem mass tag (TMT) labeling coupled with mass spectrometry. Ten samples were collected, spanning from metaphase I to metaphase II (50–140 min). Following cell breakage and trypsin digestion, peptides were labeled using 10 isobaric tandem mass tags (TMT10plex), and mixed. Phosphopeptides were enriched using Fe-NTA metal affinity chromatography and the flow-through was kept to perform shotgun proteome analysis to evaluate the protein abundance. The phosphoproteome and proteome were analyzed using tandem mass spectrometry (LC-MS/MS) data acquisition (Fig. 1A). Data were filtered to contain only singly phosphorylated peptides and only peptides with a localization probability score greater than 0.75. Two repeats were performed (Fig. EV1B). The one with the larger number of quantified phosphosites (repeat 1) was used to illustrate the results of the article. The Appendix Fig. S1 shows the results derived from repeat 2, which are similar (Dataset EV1).

To evaluate the stability of the proteome, 757 proteins were quantified in our shotgun experiment. In the phosphoproteome experiment, we identified 4095 phosphosites and quantified 2501 phosphosites with high localization probability (Fig. EV1B)

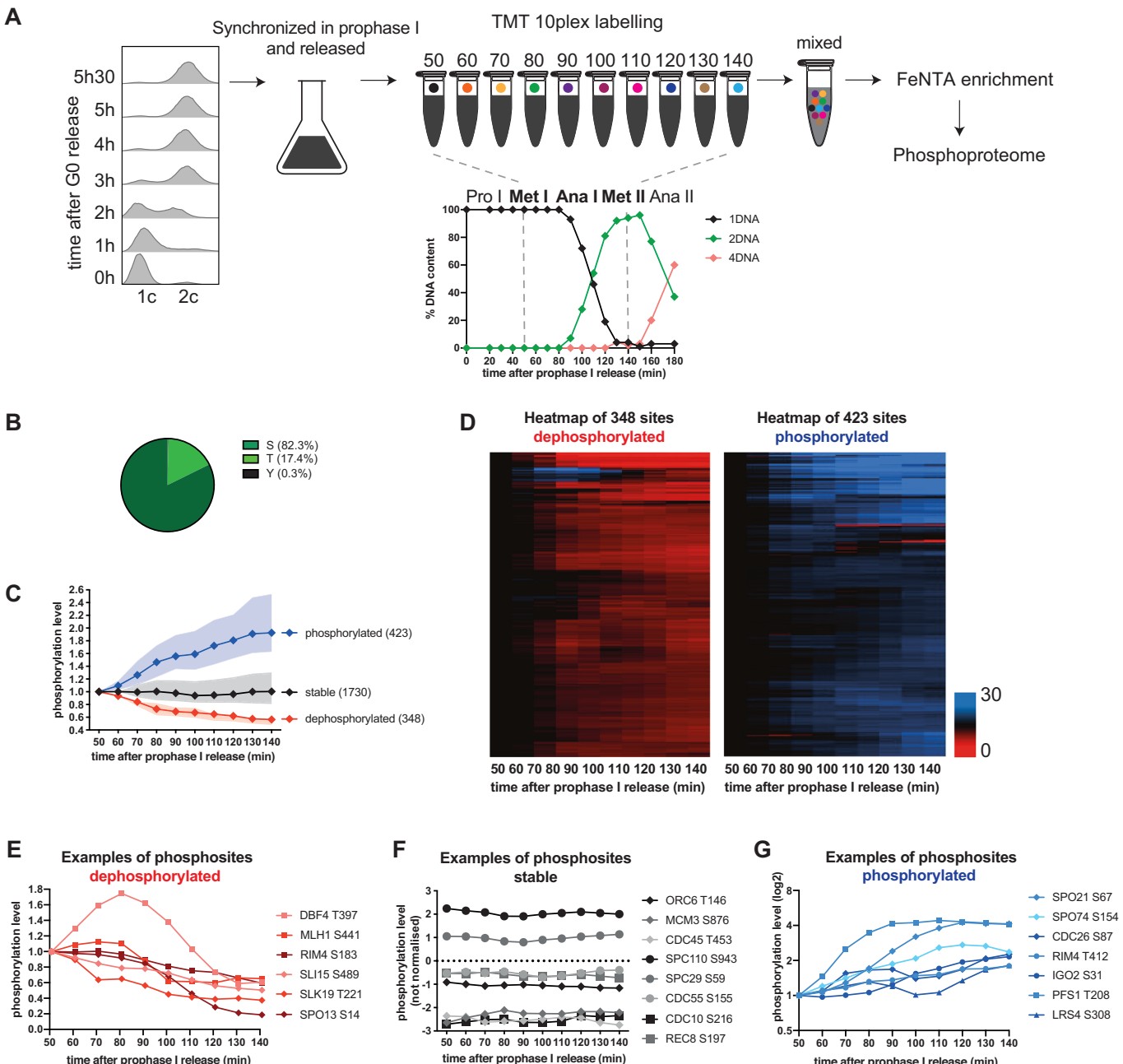

**Figure 1. The phosphorylation landscape of the MI-MII transition in budding yeast.**

(**A**) Scheme of the experiment. *GAL4.ER GAL-NDT80* strain was induced to sporulate by transfer into SPO medium at 0 h and grown at 30 °C. Synchronous cell progression in pre-meiotic S-phase was monitored by FACS analysis of DNA content. At 5h30, cells were arrested in prophase I by using the *GAL4.ER GAL-NDT80* arrest and release system. After addition of 1 μM β-estradiol, cells progress through meiosis I. One hundred cells were scored for spindle length and DNA content at each timepoint (0 to 180 min after β-estradiol addition) to determine the cell cycle phases. Samples progressing from metaphase I (50 min) to metaphase II (140 min) were used for the mass spectrometry experiment using TMT10plex labeling. 1DNA, 2DNA and 4DNA indicate the number of DNA masses observed by DAPI staining. Each isobaric mass tag is represented by a different color. After mixing, phosphopeptide enrichment and liquid chromatography-tandem mass spectrometry (LC-MS/MS) were performed. (**B**) Fractions of quantified serine (S), threonine (T), or tyrosine (Y) phosphosites. (**C**) Normalized median intensity profiles and interquartile range of the phosphosites that undergo a 1.5-fold decrease (red) or a 1.5-fold increase (blue) in phosphorylation abundance through meiosis I exit. Normalized median intensity profiles and interquartile range of the phosphosites remaining stable are in black. (**D**) Heatmap of abundance changes of dephosphorylated sites (left) and phosphorylated sites (right) during meiosis I exit. Phosphorylation abundance is normalized to the first time point (50 min). Rows were ordered by unsupervised hierarchical clustering (no k-mean). Increase in phosphorylation is visible in blue and decrease in red. (**E**) Profile plot examples of dephosphorylated phosphosites. (**F**) Profile plot examples of stable phosphosites. Note that for better visualization the phosphorylation intensity was not normalized to 1. (**G**) Profile plot examples of phosphorylated phosphosites. Phosphosite intensities are shown on a log2 scale. Source data are available online for this figure.

corresponding to 1150 unique phosphoproteins (Fig. EV1C). The overlap with the phosphoproteins is illustrated in Fig. EV1C. As previously described in mitosis, serines (82.3%) were the most preferred phosphoacceptor over threonines (17.4%). Tyrosines were rarely phosphorylated (0.3%) (Fig. 1B). To categorize phosphosite behavior, phosphorylation levels at the first time point (50 min) were normalized to 1 (i.e., 0 on a log2 scale). 1.5-fold cut-off was used to categorize phosphosites as phosphorylated or dephosphorylated during meiosis I exit. Phosphosites undergoing a 1.5-fold decrease or increase over at least two consecutive time points were classified as dephosphorylated or phosphorylated, respectively, and the others were classified as stable. As previously observed during mitotic exit, the proteome is mainly stable during meiosis I exit, too (Fig. EV1B–D) (Touati et al, 2018). Indeed, less than 3% of the proteome fluctuated in our main dataset. Consequently, most changes in the phosphoproteome are expected to be phosphorylation-dependent. The phosphoproteome is a lot more dynamic than the proteome, with similar number of phosphosites (14–17%) lost or gained phosphorylation during the time course. Still, the majority of the phosphoproteome remained stable during meiosis I exit (Fig. 1C). The number of phosphosites and the overlap between the two repeats are summarized in Fig. EV1B,E. See Appendix Fig. S1 for detailed results of repeat 2.

To globally visualize the phosphorylation dynamics during meiosis I exit, we performed unsupervised hierarchical clustering (Fig. 1D). Examples of dephosphorylated, stable or phosphorylated phosphosites on major cell cycle regulators are presented in Fig. 1E–G, respectively. Many meiosis-specific proteins such as Spo13 and Rim4 (two essential meiotic cell cycle regulators), Rec8 (a meiosis-specific component of the cohesin complex), Lrs4 (a monopolin complex subunit), Spo21 and Pfs1 (two meiosis-specific sporulation proteins required for prospore membrane formation and morphogenesis) and Spo74 (a component of the meiotic outer plaque of the spindle pole body) were identified as targets of phosphoregulation. Crucially, proteins implicated in DNA replication initiation such as Orc6, Mcm3 and Cdc45 had nearly no variation in phosphorylation abundance during the MI-MII transition, even though Ocr6 T146 is a phosphosite completely reset by dephosphorylation during mitotic exit (Fig. 1F) (Touati et al, 2019). Examples of phosphosites dephosphorylated or phosphorylated the earliest or latest are shown in Fig. EV1F–I respectively. Many proteins implicated in the synaptonemal complex are dephosphorylated before anaphase I (Fig. EV1F). Many cell cycle regulators start their dephosphorylation during anaphase I and some of them peak in phosphorylation before being dephosphorylated (Fig. EV1G,H). Finally, we identified a last cluster of proteins whose phosphorylation starts only when cells enter meiosis II (Fig. EV1I). Altogether, our dataset allows us to capture subtle changes in phosphorylation dynamics and to highlight the timing of phosphorylation abundance change during the meiotic cell cycle progression.

## Distinct phosphoregulation of substrates specific for mitotic or meiosis I exit

We previously generated a mitotic time-resolved phosphoproteome covering events from metaphase to G1, using the GALpromoter-CDC20 block and release system and taking time points every 5 min for 45 min (Touati et al, 2018). We wanted to compare how the phosphoproteome varies between mitotic and meiosis I exit. To do so, we reanalyzed the main dataset of (Touati et al, 2018) together with the new meiotic dataset (Fig. 1). At first glance, mitotic and meiosis I exit appear similar regarding phosphosite numbers, and overall dynamics. Indeed, 31% of phosphosites were either dephosphorylated or phosphorylated and 69% were stable (Fig. 2A).

How then meiosis I exit allows partial resetting of phosphorylation events to ensure chromosome segregation while avoiding DNA replication? To determine whether partial resetting of phosphorylation events is key to allowing some specific biological processes to occur while preventing others, gene ontology (GO) analysis of phosphorylation changes at meiosis I exit was used to compare them with changes at mitotic exit. Gene ontology biological process (GOBP) analysis did not highlight any particular process where the phosphorylation status of a given substrate group was reversed when comparing mitotic and meiosis I exit. However, gene ontology cellular component (GOCC) analysis yielded differences (Fig. 2B; Appendix Fig. S1). Specific meiotic cellular component categories such as the synaptonemal complex or prospore membrane had substrates mainly dephosphorylated or phosphorylated, respectively, during the MI-MII transition and as expected, were not enriched during mitotic exit. Examples of phosphosite profiles are shown in Fig. EV2A. During mitotic exit, substrates are strongly dephosphorylated at the cellular bud neck, and this is not observed during meiosis I exit (where the bud neck is absent). Finally, we found spindle pole body substrates principally phosphorylated during meiosis I exit, while they tend to be rather dephosphorylated during mitotic exit (Fig. 2B). Specific examples of spindle pole body phosphosites identified in the two datasets are shown in Fig. EV2B. This result suggests that some specific compartments have a kinase/phosphatase ratio that is different between mitotic and meiosis I exit. Altogether, this makes sense for the spindle pole body, which needs to be tightly regulated to allow a second round of chromosome segregation without interfering with the DNA replication program at the MI-MII transition. However, except for this particular case, cellular localization of phosphorylated substrates is unlikely the main driver promoting distinct events to happen between mitotic and meiosis I exit.

## Phosphorylation dynamics of mitotic kinase consensus motifs are reversed at meiosis I exit

Apart from substrates specific to mitotic or meiosis I exit as well as the spindle pole body components, our phosphoproteome landscape did not highlight specific groups of substrates phosphoregulated in a distinct manner between mitotic and meiosis I exit. Thus, we hypothesized that additional characteristics defining mitotic and meiosis I exit are found at the level of phosphomotifs. Without focusing on any substrate categories, we previously showed that sequential waves of kinase activities, the nature of the phosphoacceptor residue and the phosphomotifs surrounding the phosphosite, are programming the order of mitotic substrate phosphorylation (Touati et al, 2018). We wanted to determine whether these properties also drive specific meiotic phosphorylation events and how phosphorylation on specific motifs may differ from mitotic exit. To do so, we classified phosphosites by their kinase consensus motif signatures. Plotting the median phosphorylation timing of full Cdk (S/T)Px(K/R), Plk (D/E/N)x(S/T) and

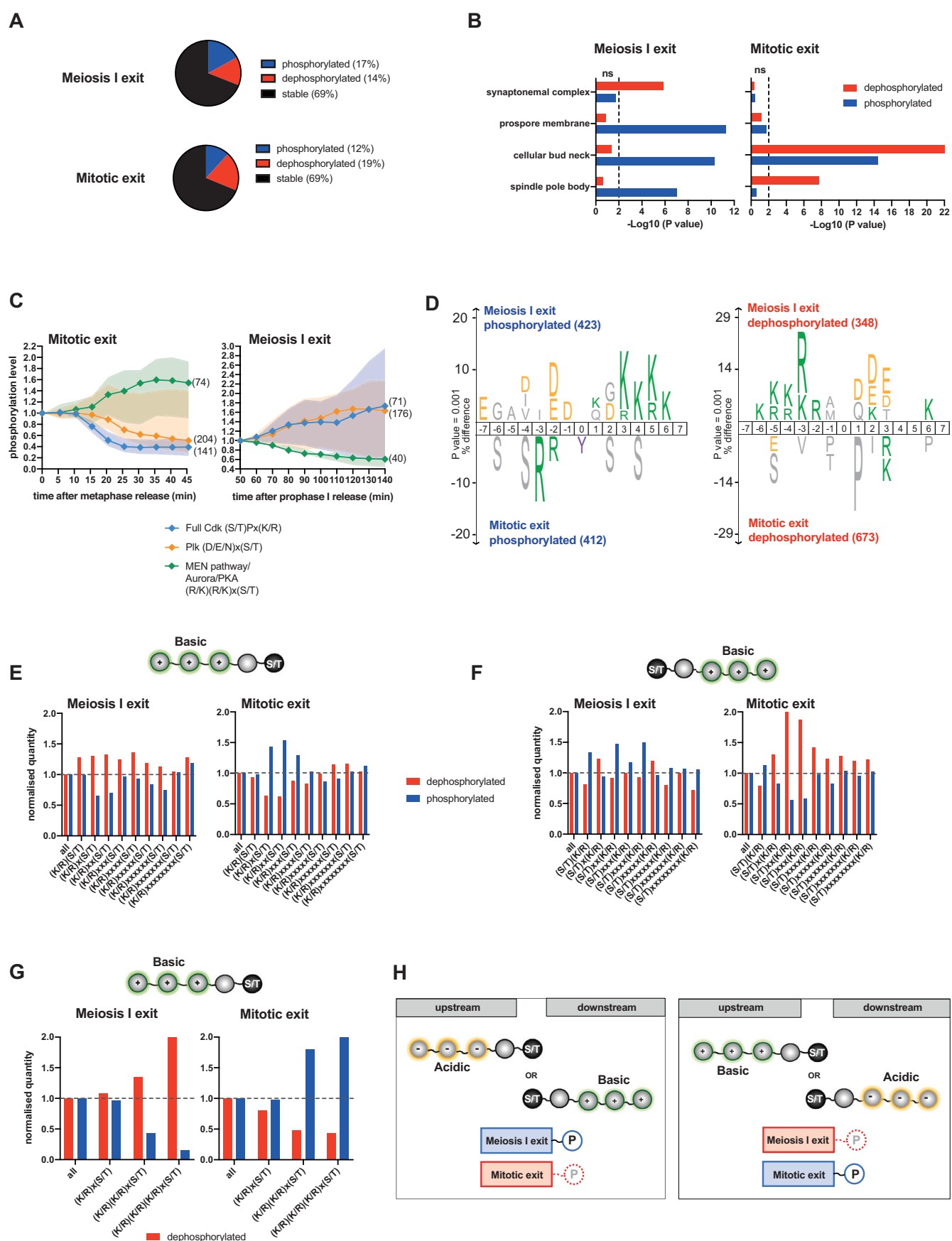

**Figure 2. Phosphorylation dynamics of mitotic kinase consensus motifs are reversed at meiosis I exit.**

(A) Fractions of phosphosites phosphorylated (blue), dephosphorylated (red), or remaining stable (black) through meiosis I exit (above) and mitotic exit (below). (B) Enrichment analysis of gene ontology cellular compartment (GOCC). −Log10 raw $p$-value is represented for the dephosphorylated proteins (red) and the phosphorylated proteins (blue) in the mitotic exit and the meiosis I exit datasets. Non-significant (ns) enrichment is represented on the left of the dashed line and significant enrichment on the right ($P < 0.05$) (Mann–Whitney $U$ Test). (C) Normalized median intensity profiles and interquartile range of the phosphosites that adhere to the three indicated kinase consensus motifs. For better visualization of the data, stable phosphosites of each category were not included in the median intensity profile. Of note (D/E/N)x(S/T) and (R/K)(R/K)x(S/T) categories exclude Cdk sites (S/T)P. (D) IceLogo motif analysis of phosphosites to reveal sequence elements enriched around phosphorylation sites. These analyses compared sites that are phosphorylated (left) and dephosphorylated (right) during meiosis I exit versus mitotic exit. The phosphorylated residue is at position 0. Larger letter size indicates increased enrichment. Percentage of difference is used as scoring method. The threshold for enrichment detection was $p = 0.001$. (E–G) Percentage of dephosphorylated and phosphorylated sites categorized by amino acid motif identity after normalization to the total amount of sites. Phosphosites with distinct basic upstream amino acids are presented in (E), with distinct basic downstream amino acids in (F), and with single, double, or triple basic upstream amino acids in (G). (H) Scheme of phosphomotifs preferentially phosphorylated and dephosphorylated during meiosis I exit and mitotic exit. Serine and threonine (S/T) residues with acidic (yellow) upstream amino acids or basic (green) downstream amino acids are preferentially phosphorylated (blue) during meiosis I exit and dephosphorylated (red) during mitotic exit (left). Serine and threonine (S/T) residues with basic (green) upstream amino acids or acidic (yellow) downstream amino acids are preferentially dephosphorylated (red) during meiosis I exit and phosphorylated (blue) during mitotic exit (right). Source data are available online for this figure.

Aurora/PKA/MEN pathway (R/K)(R/K)x(S/T) consensus sites revealed that their profiles were completely reversed between mitotic and meiosis I exit (Fig. 2C; Appendix Fig. S1).

IceLogo motif analysis is another way to reveal kinase consensus motifs enriched around the phosphosite. Again, comparison between the phosphorylated and dephosphorylated sequences during mitotic and meiosis I exit shows a reverse distribution of basic and acidic residues around the phosphosite (Fig. 2D). Moreover, this recapitulates the kinase consensus motif behavior observed in Fig. 2C. Strikingly, even if the mitotic and meiosis I exit phosphoproteomes look similar in terms of global phosphorylation and dephosphorylation distribution, phosphosites fitting these major kinase categories are clearly distinct.

## Basic and acidic phosphomotif patterns are inverted between mitotic and meiosis I exit

IceLogo motif analysis highlighted differences in the phosphorylation dynamic of precise kinase consensus motifs between mitotic exit and meiosis I exit. We also observed that the global environment around the phosphosite—basic or acidic— modify the phosphosite behavior. So we decided to take an unbiased approach and investigate how the position of basic or acidic residues around the phosphoacceptors (serine or threonine) affects whether a phosphosite is dephosphorylated or phosphorylated during mitotic and meiosis I exit. First, we checked if basic and acidic residues upstream or downstream of the phosphoresidues were equally distributed in mitosis and meiosis. For both we noticed that a basic residue (K/R) in −2 position or some acidic residues (D/E) in +1 or +2 position increases the chances of phosphorylation, suggesting that several kinases adhere to those phosphomotifs. However, phosphoresidues with basic residues (K/R) in +1–4 positions in mitosis and meiosis are less frequently phosphorylated (Fig. EV2C). The phosphoenrichment technique used may increase the affinity for some phosphomotifs over others. However, we think this is unlikely to generate a bias in our analysis, as all our peptides were mixed before being phosphoenriched on the same column. Also, we do not just determine whether a phosphosite is phosphorylated, but rather if phosphorylation on a specific phosphosite increases or decreases overtime.

After normalization according to the number of each phospho-motif identified in the two datasets we analyzed if the position of basic or acidic residues impacts the behavior of a phosphosite during exit

(i.e., being dephosphorylated or phosphorylated) (Fig. 2E–G, EV2D–G and Appendix Fig. S1). We found that the fate of phosphosites during exit from mitosis and meiosis I is reversed and depends on the distribution of basic and acidic residues up- and downstream. For example, basic residues located upstream of the phosphosite favor dephosphorylation during meiosis I exit while they promote phosphorylation during mitotic exit (Fig. 2E). On the contrary, basic residues downstream in position 3–5 promote phosphorylation during meiosis I exit and dephosphorylation during mitotic exit (Fig. 2F). Acidic residues downstream and upstream also show a reversed pattern in promoting phosphorylation or dephosphorylation when comparing mitotic and meiotic exit (Fig. EV2D). We also noticed that the number of basic or acidic residues affects the phosphorylation status of the phosphosite (Fig. 2G and EV2E,G). For example, the more basic residues upstream of the phosphosite, the more likely the phosphosite was to be dephosphorylated at meiosis I exit and phosphorylated at mitotic exit (Fig. 2G). This parameter also affects the kinetics of phosphosite dephosphorylation. The more basic residues are present upstream of the phosphosite, the faster the phosphorylation of the phosphosite decreased during meiosis I exit (Fig. EV2H). Also, the presence of several basic residues downstream of the phosphosite during mitotic exit sped up dephosphorylation timing and range (Fig. EV2I). Altogether, the global distribution of basic and acidic amino acids around the phosphosites influence its chance to be differently regulated between mitotic and meiotic exit. The results of Fig. 2E–G and EV2D–I are summarized in the scheme in Fig. 2H. The charge switch of consensus sites when comparing mitotic and meiosis I exit is very intriguing and suggests that kinase and phosphatase profiles are fundamentally different.

## Full Cdk sites dephosphorylated during mitotic exit are mostly stably phosphorylated during meiosis I exit

Dephosphorylations are critical to reset the cell cycle during mitotic exit, but whether a partial reset also occurs during meiosis I exit remained still elusive. To answer this question, we performed a direct comparison of phosphosite behavior during mitotic and meiosis I exit. To do so, we analyzed the 727 phosphosites that are in common between the two datasets (Fig. 3A). The Sankey diagram shows that the majority of phosphosites that remain stable during mitotic exit also remain stable during meiosis I exit. Next, we focused our attention on phosphosites that are reset by dephosphorylation during mitotic exit and uncovered three categories of meiotic

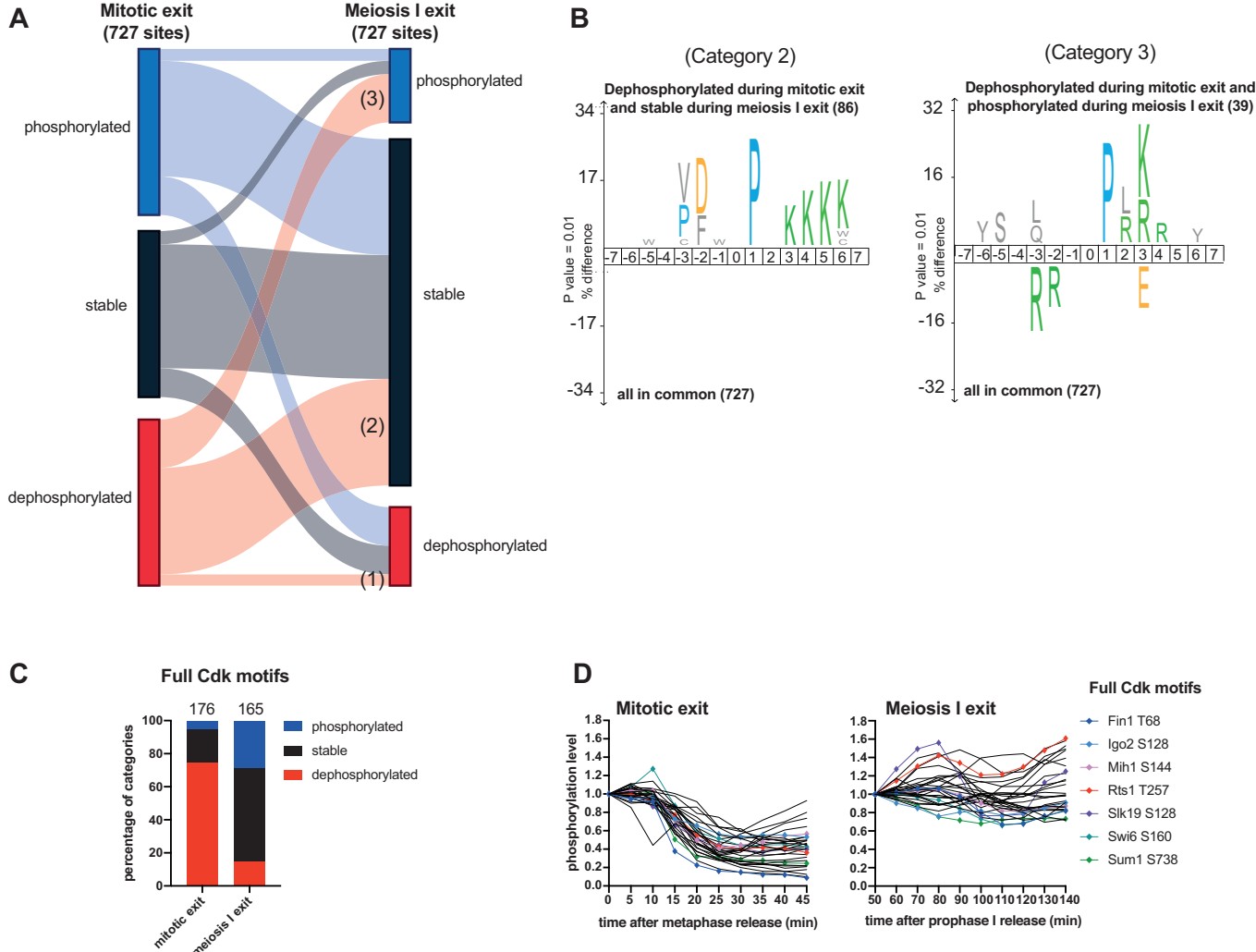

**Figure 3. Full Cdk sites dephosphorylated during mitotic exit are mostly stably phosphorylated during meiosis I exit.**

(A) Sankey diagram showing phosphosites in common between the mitotic exit dataset and the meiosis I exit dataset (727 sites in total). For better visualization of the data, phosphorylated (blue), stable (black) and dephosphorylated (red) categories in the mitotic exit dataset are normalized to 100%. The size of the connecting lines is proportional to the number of phosphosites in each category in the mitotic exit dataset. The table Fig. EV3A shows the number of phosphosites in each category. (1) Shows the category of phosphosites dephosphorylated both during mitotic exit and meiosis I exit. (2) Shows the category of phosphosites dephosphorylated during mitotic exit and stable during meiosis I exit. (3) Shows the category of phosphosites dephosphorylated during mitotic exit and phosphorylated during meiosis I exit. (B) IceLogo motif analysis of phosphosites to reveal sequence elements enriched around phosphorylation sites. These analyses compared the 86 sites (2) (left) and the 39 sites (3) (right) of the Sankley diagram in Fig. 3A with all the phosphosites in common between the mitotic exit and the meiosis I exit dataset. The phosphorylated residue is at position 0. Larger letter size on the top indicates increased enrichment. Percentage of difference is used as scoring method. The threshold for enrichment detection was $C = 0.001$. (C) Percentage of the full Cdk sites (S/T)Px(K/R) in each category. (D) Profile plot of the 31 full Cdk sites in common between the mitotic exit dataset (left) and the meiosis I exit dataset (right) that are dephosphorylated during mitotic exit and remain stable during meiosis I exit. Profiles of essential cell cycle regulators are highlighted in colors. Source data are available online for this figure.

phosphosubstrates with distinct phosphorylation dynamics. Category (1) includes only 9 phosphosites that were dephosphorylated during both mitotic and meiosis I exit (Fig. EV3A). Their profiles are shown Fig. EV3B. The majority of the phosphosites dephosphorylated during mitotic exit were either stable (category (2)) or phosphorylated (category (3)) during meiosis I exit (Fig. 3A and EV3A). IceLogo motif analysis was performed on categories (2) and (3) and the full Cdk consensus motif emerged (Fig. 3B). Indeed, while full Cdk motifs are efficiently dephosphorylated during mitotic exit, they remain stable or even increase in phosphorylation during meiosis I exit (Fig. 3C and EV3C). We plotted the profiles of the 31 full Cdk motifs

that are dephosphorylated during mitotic exit and that are also present in the meiosis I exit dataset, highlighting well-known cell cycle regulators. Some of the sites undergo subtle variations at the MI-MII transition but overall, they remain stably phosphorylated during meiosis I exit (Fig. 3D; Appendix Fig. S1). The list of the the full Cdk site dynamics during mitotic exit, and meiosis I exit (repeat 1 and 2) is available in Dataset EV1. Altogether, our results demonstrate that phosphorylations by the master Cdk cell cycle regulator are reset during mitotic exit, but not during meiosis I exit, thus highlighting a key difference in protein phosphorylation landscape during mitosis and meiosis I exit.

## Serine and threonine phosphoresidues on full Cdk sites show different phosphorylation dynamics

We next investigated whether the presence of basic or acidic residues around phosphoacceptors impacts substrate phosphorylation, considering a proline in the +1 position (Fig. 4A). This

revealed a difference in the behavior of serines and threonines on full Cdk motifs. Indeed, while SPx(K/R) and TPx(K/R) are equally dephosphorylated during mitotic exit, they do not follow the same pattern during meiosis I exit. TPx(K/R) are strongly phosphorylated during the MI-MII transition and Western Blot analysis using a specific TPx(K/R) antibody confirmed this tendency (Fig. 4B,C;

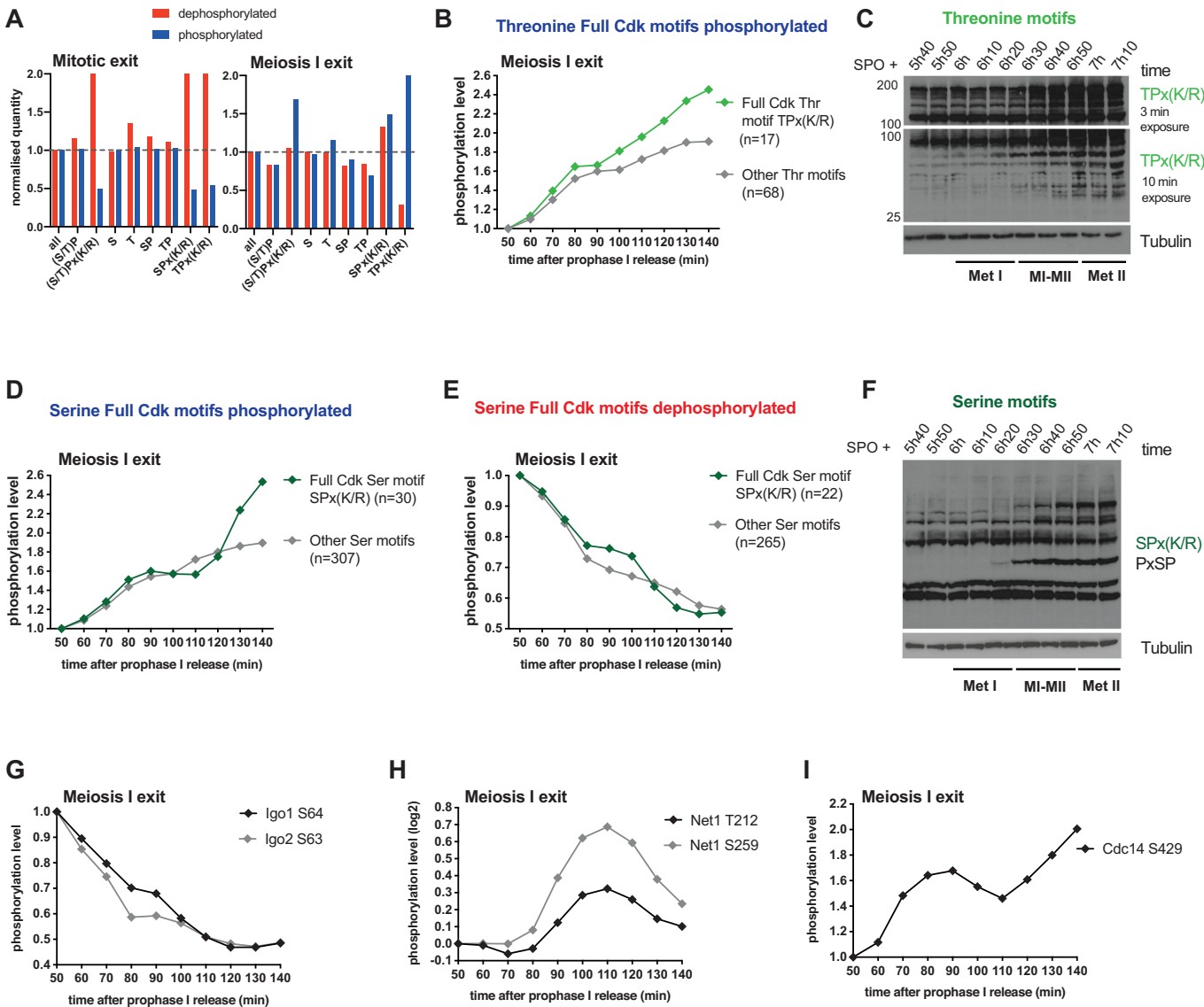

**Figure 4. Serine and threonine phosphoresidues on full Cdk sites show different phosphorylation dynamics.**

(A) Percentage of dephosphorylated (red) and phosphorylated (blue) sites categorized by amino acid motif identity after normalization to total amount of sites. (B) Normalized median intensity profiles of sites phosphorylated on the full Cdk threonine motifs TPx(K/R) (light green) or the other threonine sites (gray) during meiosis I exit. (C) Protein extracts were prepared at the indicated times from *GAL4.ER GAL-NDT80* strains and processed for Western blotting. Cells were induced to sporulate by transfer into SPO medium at 30 °C. 1 µM β-estradiol was added at 5 h to allow prophase I release. Proteins were labeled with the indicated antibody. Time after release into SPO medium and the cell cycle stages are indicated. Met I: Metaphase I. MI-MII: MI-MII transition. Met II: Metaphase II. Exposure time is indicated. One representative experiment from three independent experiments is shown. (D) Normalized median intensity profiles of sites phosphorylated on the full Cdk serine motifs SPx(K/R) (dark green) or the other serine sites (gray) during meiosis I exit. (E) Normalized median intensity profiles of the sites dephosphorylated on full Cdk serine motifs SPx(K/R) (dark green) or the other serine sites (gray) during meiosis I exit. (F) Protein extracts were prepared at the indicated times from *GAL4.ER GAL-NDT80* strains and processed for Western blotting. Cells were induced to sporulate by transfer into SPO medium at 30 °C. 1 µM β-estradiol was added at 5 h to allow prophase I release. Proteins were labeled with the indicated antibody (recognizing SPx(K/R) and PxSP phosphomotifs). Time after release into SPO medium and the cell cycle stages are indicated. Met I: Metaphase I. MI-MII: MI-MII transition. Met II: Metaphase II. One representative experiment from three independent experiments is shown. (G) Profile plot of Igo1 S64 and Igo2 S63 phosphorylation profiles during meiosis I exit. (H) Profile plot of Net1 T212 and S259 phosphorylation profiles during meiosis I exit. (I) Profile plot of Cdc14 S429 phosphorylation profile during meiosis I exit. Source data are available online for this figure.

Appendix Fig. S1). However, SPx(K/R) were either phosphorylated or dephosphorylated during meiosis I exit, as confirmed by Western Blot (Fig. 4D–F; Appendix Fig. S1). Most importantly, the phosphorylation of full Cdk motifs reached a plateau between 90 and 110 min after prophase I release, a process not observed for any other kinase motifs containing a serine residue (Fig. 4D). Since this timing corresponds to the MI-MII transition, our results indicate that the phosphorylation of full Cdk motifs behaves differently between mitosis and meiosis I exit, a process that is in part mediated by the identity of the phosphoacceptors. Notably, serine residues are dephosphorylated while threonine are not, suggesting a phosphosite preference for Cdk or the absence of a specific phosphatase with a threonine preference.

### Time-resolved phosphoproteome analysis reveals theoretical phosphatase activity profiles

Phosphatases also harbor substrate specificity. In yeast and mammals, PP2A$^{Cdc55}$ and PP1 were shown to preferentially dephosphorylate threonine motifs while Cdc14 prefers full serine Cdk sites (Baro et al, 2018; Bremmer et al, 2012; Godfrey et al, 2017; Hoermann et al, 2020; Kruse et al, 2020; Kuilman et al, 2015; Powers and Hall, 2017; Touati et al, 2019). We previously showed that Cdc14, PP2A$^{Cdc55}$ and PP2A$^{Rts1}$ cooperate to promote substrate dephosphorylation during mitotic exit (Touati et al, 2019). The modulation of phosphatase activity during meiosis I and II could be the reason why full Cdk threonine motifs are not dephosphorylated, so we decided to search for a read-out of their activities in our dataset. Concerning PP2A$^{Cdc55}$, we looked at the phosphorylation status of Igo1 and Igo2 (Arp19/ENSA in mammals) which inhibit PP2A$^{Cdc55}$ when they are phosphorylated on serines S63 and S64, respectively (Juanes et al, 2013). We found that Igo1/2 were continuously dephosphorylated from metaphase I to metaphase II (Fig. 4G; Appendix Fig. S1). These dephosphorylations are supposed to limit the interaction between Igo1/2 and PP2A$^{Cdc55}$ suggesting that PP2A$^{Cdc55}$ becomes active at MI-MII transition and remains active until meiosis II. PP2A$^{Cdc55}$ is known to preferentially target minimal Cdk and Plk motifs, which can explain why only full Cdk motifs remain strongly phosphorylated on threonine.

We then turned our attention to Net1, the inhibitor of Cdc14, which sequesters Cdc14 in the nucleolus. In mitosis, Cdc14 shows a two-step release, first from the nucleolus to the nucleus and second from the nucleus to the cytoplasm; however, in meiosis, at the MI-MII transition, Cdc14 is not fully released into the cytoplasm. While the two pathways responsible for full Cdc14 release are active in mitosis—the FEAR and MEN pathway— only the FEAR pathway is active at MI-MII transition (Buonomo et al, 2003; Kamieniecki et al, 2005; Marston et al, 2003; Pablo-Hernando et al, 2007). When Net1 is phosphorylated on specific sites (including T212 and S259) in mitosis, Cdc14 is released and activated (Azzam et al, 2004; Queralt and Uhlmann, 2008). We found that Net1 T212 and S259 were phosphorylated in anaphase I and dephosphorylated in metaphase II suggesting that Cdc14 is reactivated at the MI-MII transition, but inactive in metaphase II (Fig. 4H; Appendix Fig. S1). Phosphorylation of Cdc14 on S429 is thought to inhibit its phosphatase activity (Li et al, 2014). Importantly, Cdc14 S429 phosphorylation decreases completely during mitotic exit but only partially at the MI-MII transition, before increasing again in meiosis II. In our meiotic dataset, the majority of SPx(K/R) is

poorly dephosphorylated, indicating that indeed, Cdc14 is not fully active (Fig. 4I; Appendix Fig. S1) (Touati et al, 2019; Touati et al, 2018). Thus, when comparing the mitotic and meiosis I exit datasets, the phosphorylation balance is ascending during meiosis I exit as Cdk activity is only partially reduced (Carlile and Amon, 2008) and Cdc14 is only partially active. Thus, full Cdk motifs are barely dephosphorylated at meiosis I exit, especially the TPx(K/R), which are less favored by Cdc14.

### Phosphorylation landscape at meiosis I exit after complete drop in Cdk activity

Cdk directly regulates the activity of downstream kinases and phosphatases but also provides priming phosphorylations on substrates allowing other kinases to dock and phosphorylate phosphosites at proximity. At this stage, we were suspecting that the opposite amino acid patterns between the mitotic and meiotic cell cycle exit were due to the fact that Cdk activity clearly drops during mitotic exit and only partially during meiosis I exit (Carlile and Amon, 2008). To test this hypothesis, we decided to mimic a mitotic exit-like situation by strongly decreasing Cdk activity just before anaphase I, and study the consequences on the phospho-proteome. If our hypothesis was correct, the complete loss of Cdk activity should result in a phosphorylation landscape similar to mitotic exit.

Again, the GALpromoter-NDT80 block-release system was used, but this time in combination with the *cdc28-as1* mutant as the sole Cdk, to be able to specifically and immediately inhibit Cdk activity after the addition of non-hydrolysable ATP analogues (Bishop et al, 2000). In our study, we used a high concentration of the 3-MB-PP1 analogue for complete inhibition. FACS analysis showed that cells went synchronously through the pre-meiotic cell cycle stages (Fig. 5A). At 5 h, β-estradiol was added, and time points were collected at 10 min intervals to confirm the synchrony of meiotic progression by immunofluorescence (Fig. 5B). 60 min after β-estradiol addition, cells had reached metaphase I and the first time point (60 min) was collected. The culture was immediately split in two and 3-MB-PP1 at 10 μM final concentration was added to one of the cultures. Time points were collected every 5 min during 25 min. We monitored the decrease in Cdk activity in metaphase I by Western Blot, using the antibody against SPx(K/R) phosphomotifs. As expected, we observed the disappearance of prominent bands in presence of 3-MB-PP1, confirming that Cdk activity is decreased less than 5 min after treatment (Fig. 5C). Samples were collected and processed for LC-MS/MS after TMT11plex labeling and phosphopeptide enrichment (Fig. 5A, Dataset EV1). DNA immunofluorescence showed that the three mass spectrometry analysis repeats behaved in the same way in control and after drug addition. Note that one of the repeats (repeat B) managed to perform anaphase I more efficiently than the two other repeats, but still with a delay of 1 h compared to the control (Fig. EV4A).

### Full Cdk motifs react fast to inhibition of Cdk activity in meiosis I

We quantified 1529, 1452 and 1262 phosphosites with high localization probability in repeat A, B and C, respectively. The number of phosphosites quantified and the overlap between the

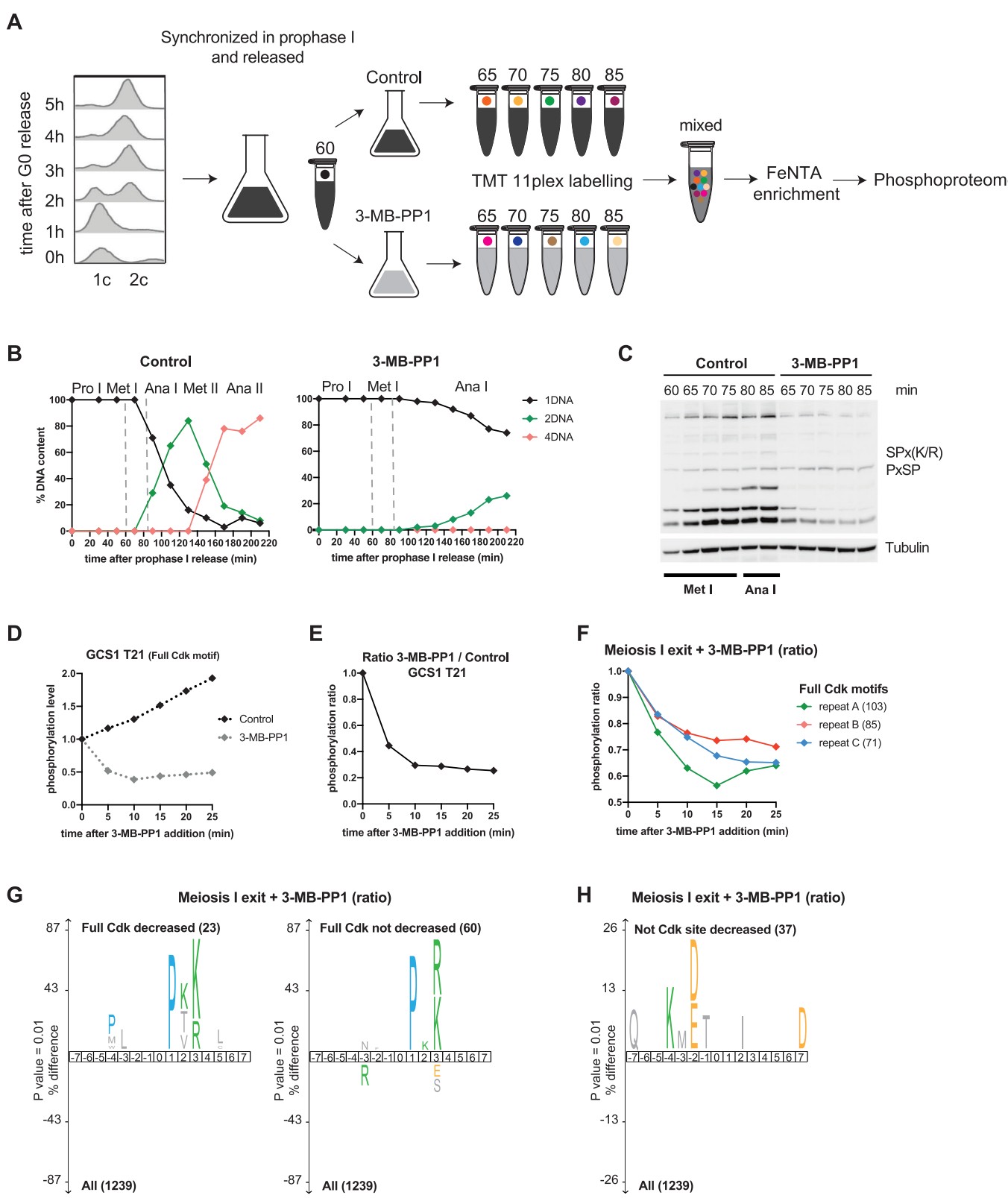

three repeats are summarized in Fig. EV4B. Repeat A counted the highest number of phosphopeptides and was used to illustrate the profile plots presented, time 0 corresponds to 3-MB-PP1 addition. An example of the behavior of a full Cdk motif, GCS1 T21, is shown in Fig. 5D. We observed that GCS1 T21 is normally phosphorylated during meiosis I exit but in the presence of 3-MB-PP1, T21 phosphorylation decreased with an exponential decay. The ratio of 3-MB-PP1/Control was calculated at each time point to

**Figure 5. Phosphorylation landscape at meiosis I exit after complete drop in Cdk activity.**

(A) *GAL4.ER GAL-NDT80 cdc28-as1 (F88G)* strain was induced to sporulate by transfer into SPO medium at 0 h and grown at 30 °C. Synchronous cell progression through pre-meiotic S-phase was monitored by FACS analysis of DNA content. At 5 h, cells were arrested in prophase I by using the *GAL4.ER GAL-NDT80* arrest and release system. After the addition of 1 µM β-estradiol, cells progress through meiosis I. The culture progressing into metaphase I was divided in two flasks and 10 µM of 3-MB-PP1 or DMSO was added directly after collecting the first time point (60 min). Time points were collected every 5 min during 25 min and processed for MS by using TMT11plex labeling. Each isobaric mass tag is represented by a different color. After mixing, phosphopeptide enrichment and liquid chromatography-tandem mass spectrometry (LC-MS/MS) were performed. (B) One hundred cells were scored for DNA content at each timepoint (0–220 min after β-estradiol addition) in absence or presence of 3-MB-PP1 to determine the cell cycle phases. 1DNA, 2DNA and 4DNA indicate the number of DNA masses observed by DAPI staining. (C) Protein extracts were prepared at the indicated times from *GAL4.ER GAL-NDT80 cdc28-as1 (F88G)* strains and processed for Western blotting. Cells were induced to sporulate by transfer into SPO medium at 30 °C. 1 µM β-estradiol was added at 5 h to allow prophase I release. Timepoint 60 indicates the moment when 10 µM 3-MB-PP1 was added (60 min after β-estradiol addition). Proteins were labeled with the indicated antibody (recognizing SPx(K/R) and PxSP phosphomotifs). One representative experiment from three independent experiments is shown. (D) Profile plot example of phosphosite behavior in absence (Control) or presence of 3-MB-PP1. Note that for better visualization the phosphorylation intensity was not normalized to 1. (E) Ratio of 3-MB-PP1/Control of Fig. 5D was calculated at each time point. Ratio values are plotted. (F) Median intensity profile of sites phosphorylated on the full Cdk motifs in the 3 repeats. Ratio values are plotted. (G) IceLogo motif analysis of full Cdk sites decreasing fast (left) and the other full Cdk sites (right) after addition of 3MB-PP1. Percentage of difference is used as scoring method. The threshold for enrichment detection was $p = 0.01$. Only phosphosites following the same behavior in at least 2 datasets are presented. (H) IceLogo motif analysis of non Cdk sites decreased after addition of 3MB-PP1. Percentage of difference is used as scoring method. The threshold for enrichment detection was $p = 0.01$. Only phosphosites following the same behavior in at least 2 datasets are presented. Source data are available online for this figure.

identify which phosphosites were affected by the decrease in Cdk activity. Figure 5E illustrates the ratio of the curves in Fig. 5D. After normalization of the 3 repeats, a 1.5-fold cut off was used to categorize phosphosites as increased or decreased after 3-MB-PP1 addition, the percentage of each category is presented for each repeat in Fig. EV4C.

In each dataset, we observed that the full Cdk motif is enriched (as well as the Plk motif in repeat C) (Fig. EV4D). Phosphosites that decrease the most instantaneous are expected to be direct Cdk substrates while the ones that decrease with a delay are expected to be indirect substrates (Kanshin et al, 2017; Swaffer et al, 2016; Swaffer et al, 2018). Plotting the median phosphorylation timing of full Cdk motifs in each dataset showed that they all behave as expected (i.e., when Cdk activity is decreased, full Cdk phosphosites also decrease) (Fig. 5F). Repeat A had the highest range of decrease and repeat B the lowest (Fig. 5F), which can explain why cells managed to perform anaphase I more easily in repeat B of Fig. EV4A. Only phosphosites following the same behavior in at least two datasets were used for global IceLogo motif analysis to determine which category of phosphomotifs is most affected by Cdk decrease. We noticed that (S/T)PxK sites were more efficiently dephosphorylated than (S/T)PxR (Fig. 5G). This result is consistent with in vitro phosphopeptide analysis (Suzuki et al, 2015). Thus, the presence of a K or R downstream could be a strategy employed by the cell to modulate phosphorylation levels of some substrates. During mitotic exit, minimal Cdk motifs were poorly dephosphorylated compared to their full Cdk motif counterparts (Fig. 3C and EV4E,F). However, they still appeared to be dephosphorylated more efficiently during mitosis than at meiosis I exit. After 3-MB-PP1 treatment, minimal Cdk motifs decreased to a similar extent at meiosis I exit as during mitotic exit (Fig. EV4F). Thus, minimal Cdk motifs are suboptimal Cdk targets that still follow Cdk activity fluctuations in meiosis.

Finally, we turned our attention to sites other than Cdk and found to decrease upon addition of the inhibitor. We observed the appearance of the Plk motif, suggesting that the decrease in Cdk activity also indirectly affected other kinases (Fig. 5H). Overall, we have shown that a drop in Cdk activity at the MI-MII transition leads to a fast dephosphorylation of full Cdk motifs, especially SPxK (Fig. 5G). However, not all full Cdk motifs dropped straight away, suggesting that the combined action of phosphatases is

important to completely dephosphorylate some full Cdk motifs. Minimal Cdk motifs are much less affected by the drop in Cdk activity, but other motifs, including the Plk motif, are very much affected. This suggests that upon a complete drop in Cdk activity, other kinases and phosphatases are inactivated or reactivated, respectively, a few minutes after the drop. Overall, full Cdk and Plk sites behave similarly at meiosis I exit and at mitotic exit when Cdk is inhibited by 3-MB-PP1.

## Decreasing Cdk activity during meiosis I exit converts the meiotic to a mitotic phosphoproteome landscape

Our data suggest that decreasing Cdk activity at meiosis I exit restores a mitotic exit-like state in terms of phosphorylation patterns as seen by the behavior of full Cdk and Plk motifs in 3-MB-PP1-treated cells, because they are no longer inverted between mitosis and meiosis I exit (Fig. 2C). However, whether this difference is indeed due to the Cdk threshold at exit remains unclear. To address this question, we plotted the median phosphorylation timing of full Cdk (S/T)Px(K/R), Plk (D/E/N) x(S/T) and Aurora/PKA/MEN pathway (R/K)(R/K)x(S/T) consensus sites in the presence of 3-MB-PP1. Strikingly, the meiotic phosphorylation patterns were now back to what we had observed during mitotic exit (Figs. 2C and 6A). This regulation also takes place on single proteins as illustrated with Acm1, a pseudosubstrate inhibitor of the APC/C, whose phosphorylation dynamics at 7 phosphosites were inverted upon 3-MB-PP1 addition (Fig. 6B). Thus, the meiotic cell cycle has the ability to switch from a meiotic to a mitotic-exit phosphorylation landscape when Cdk is fully inhibited. Most importantly, a precise Cdk threshold has to be maintained to implement the phosphoproteome landscape during the MI-MII transition.

## Certain phosphorylation patterns at meiosis I exit remain meiosis-specific despite Cdk inhibition

Not surprisingly, we do not recapitulate all phosphorylation patterns observed during mitotic exit simply by eliminating Cdk activity at exit from meiosis I. For example, we have previously shown in mitosis that different phosphorylation patterns depend on the nature of the residues and are under the control of phosphatase

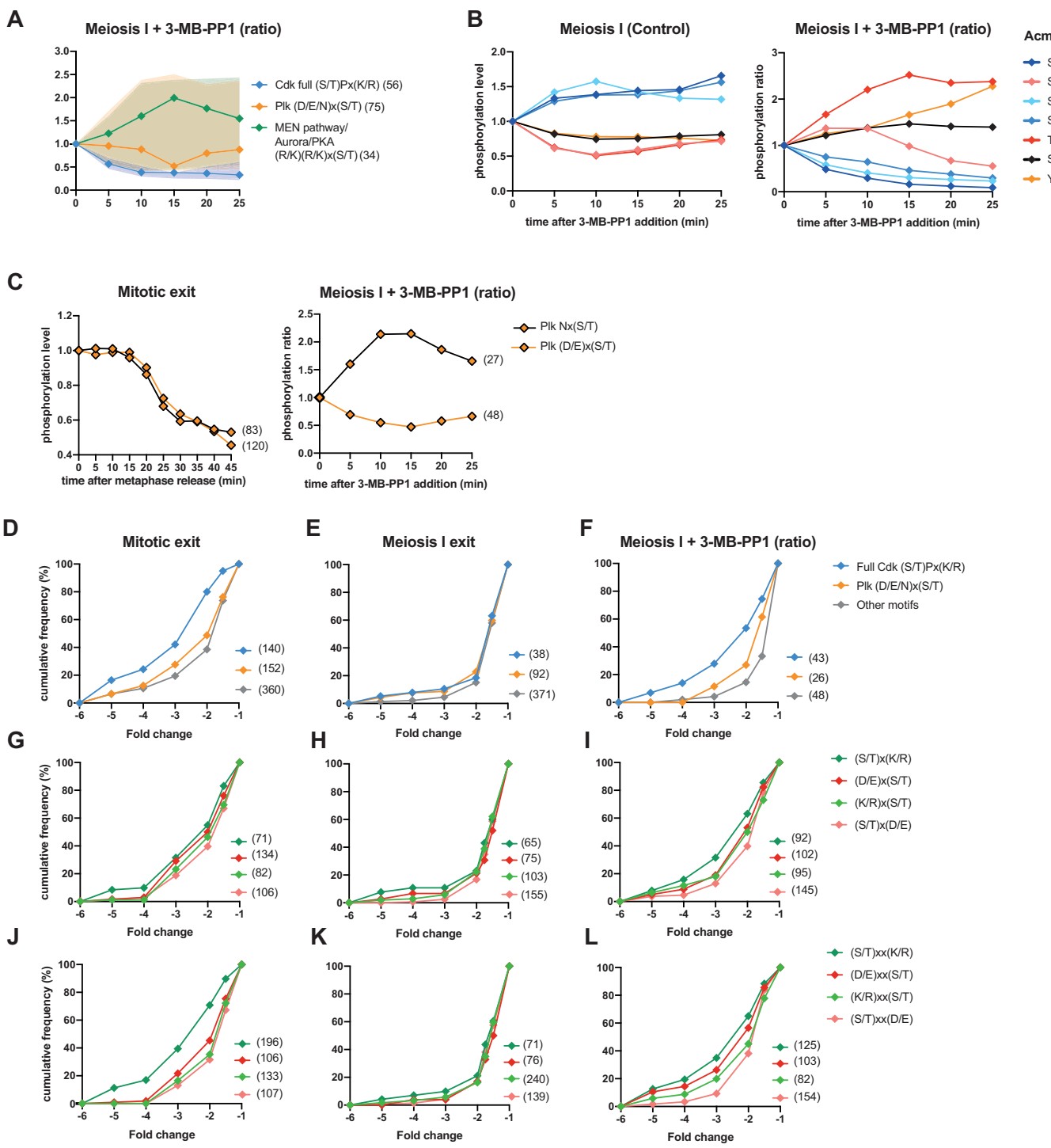

**Figure 6.  Decreasing Cdk activity during meiosis I exit converts the meiotic to a mitotic phosphoproteome landscape.**

(A) Normalized median intensity profiles and interquartile range of the phosphosites that adhere to the three indicated kinase consensus motifs. For better visualization of the data, stable phosphosites of each category were not included in the median intensity profile. Of note (D/E/N)x(S/T) and (R/K)(R/K)x(S/T) categories exclude Cdk sites (S/T)P. (B) Profile plot example of Acm1 phosphosites without (left) or with(right) treatment of 3-MB-PP1. Note that the phosphorylation ratio is presented (right). (C) Profile plot of the different Plk motifs Nx(S/T) and (DE)x(S/T) sites in the mitotic exit dataset (left) and the meiosis I exit dataset (right). (D) Cumulative frequency graph of kinase consensus motifs decreased at the indicated fold change. For the meiosis I ratio, phosphosites following the same behavior in at least 2 datasets are presented. (E–L) Cumulative frequency graph of indicated motifs decreased at the indicated fold change. For the meiosis I ratio, phosphosites of repeat A are presented. Source data are available online for this figure.

specificity, which constantly opposes kinase phosphorylation. For instance, SP phosphosites are phosphorylated earlier than TP sites in the mitotic cell cycle (Godfrey et al, 2017). Furthermore, SPx(K/R) sites are dephosphorylated before TPx(K/R) sites during mitotic exit (Touati et al, 2019). By simply reducing Cdk activity in meiosis I, these distinct dephosphorylation patterns of Cdk sites at serine and threonine were not observed (Fig. EV5A,B). Furthermore, Plk consensus sites are also distinctly regulated by phosphorylation. During mitotic exit, both Nx(S/T) and (D/E)x(S/T) sites are dephosphorylated with similar kinetics. However, during meiosis I exit, (D/E)x(S/T) were mainly dephosphorylated while Nx(S/T) phosphorylation increased upon Cdk inhibition (Fig. 6C). Thus, in meiosis I, Nx(S/T) appears to be resistant to the artificial drop in Cdk activity suggesting that Plk can keep Nx(S/T) sites phosphorylated, while it cannot counteract (D/E)x(S/T) phosphatases that dephosphorylate (D/E)x(S/T) at the same time. Alternatively, we cannot exclude that not Plk but another kinase, possibly meiosis II-specific, recognises these Nx(S/T) sites independently of Cdk activity.

## Cdk activity determines not when, but which phosphomotifs are dephosphorylated in meiosis I

It was shown previously in mitosis that the right timing of substrate phosphorylation on specific motifs relies on the balance between different waves of kinase activity, counterbalanced by phosphatase activity (Touati et al, 2019; Touati et al, 2018). Figure 6D–L shows the fold change of specific phosphomotif dephosphorylation. The temporal order during mitotic exit is recapitulated in Fig. 6D. Full Cdk sites are dephosphorylated first, followed by Plk sites and then other sites. However, no temporal order of dephosphorylation according to kinase phosphomotifs during meiosis I exit was observed; indeed, the phosphomotifs overlap (Fig. 6E). Moreover, unlike mitotic exit, the different waves of phosphosite dephosphorylation observed in Fig. EV1F,H do not correlate with any specific phosphomotifs (Fig. 6E). We asked whether artificially decreasing Cdk activity would restore these mitotic phosphomotif waves. For this purpose, we analyzed phosphorylation patterns and composition of consensus phosphosites for Cdk, Plk and other kinases. As suspected, the kinase consensus waves were re-created upon 3-MB-PP1 addition (Fig. 6F). The same results were observed when we looked at basic and acidic residues upstream and downstream of the phosphosite at −2 and +2 positions (Fig. 6G–I) and at −3 and +3 positions (Fig. 6J–L). Hence, during meiosis I exit, the cell has the plasticity to recapitulate the temporal phosphorylation patterns observed during mitotic exit. However, this strategy is not employed to order substrate dephosphorylation during the MI-MII transition to orchestrate the meiotic cell cycle.

## Discussion

Phosphorylation and dephosphorylation events shape the phosphoproteome landscape, but how these overall changes relate to function remains largely unknown. Recent advances in the mass spectrometry field have provided insights by allowing time-resolved monitoring of changes in phosphosite dynamics during cell cycle progression. Most of the phosphorylation events are stable and only 20–30% of them are dynamic in budding yeast as well as in mammals (Godfrey et al, 2017; McCloy et al, 2015; Touati et al, 2018; Touati and Uhlmann, 2018). In yeast, insights into the substrate preferences of mitotic kinases and phosphatases have provided a better understanding of the global regulation of key cell cycle transitions. However, a time-resolved picture of the meiotic phosphoproteome landscape compared to mitosis was still missing to understand how a partial reset of the cell cycle occurs.

Two models can explain how the cell cycle is ordered, the quantitative model based on Cdk activity levels and the qualitative model based on different cyclin specificities. Both models describe how a given cell cycle protein is phosphorylated and dephosphorylated at the correct cell cycle stage. The quantitative model also rationalizes the ordering of phosphosite dephosphorylation during budding yeast mitotic exit where the cell cycle is reset by a progressive decrease in Cdk kinase activity and a concomitant increase in Cdc14 phosphatase activity. This specific balance between kinases and phosphatases allows the substrates to be dephosphorylated in the correct order to return to G1 and to re-license clusters of proteins in preparation for another cycle (Bouchoux and Uhlmann, 2011; Kuilman et al, 2015; Touati et al, 2019; Touati et al, 2018). During mitotic exit, we have previously shown that a sequential pattern of kinase consensus motif dephosphorylation, starting with full Cdk motifs, shapes the phosphoproteome landscape (Touati et al, 2018). In fission yeast, it has been shown that this depends on Cdk activity levels (Swaffer et al, 2018), and we showed in budding yeast that full Cdk sites are dephosphorylated faster and with a higher fold change than Plk sites. The presence of basic or acidic residues around the phosphosites also modulates the extent of motif phosphorylation. However, we do not recapitulate these patterns in meiosis. During meiosis I exit, we believe that Cdk activity drops only partially, in agreement with Carlile and Amon (Carlile and Amon, 2008), and Cdc14 not being fully active, in agreement with other studies (Buonomo et al, 2003; Kamieniecki et al, 2005; Marston et al, 2003; Pablo-Hernando et al, 2007). Thus, the global range of phosphosite dephosphorylation at the MI-MII transition may be too low to generate successive waves of kinase consensus motif dephosphorylation due to the drop in Cdk activity levels, such as at mitotic exit. However, when Cdk activity completely drops during meiosis I due to artificial inhibition, the successive phosphomotif dephosphorylation waves of mitotic exit are restored, suggesting that the meiotic cell has the capacity to use the quantitative model to order substrate dephosphorylation at the MI-MII transition, but does not use this strategy.

Even if Cdk activity levels do not directly influence the relative timing of phosphomotif dephosphorylation during meiosis I exit, we show that Cdk activity controls which kind of phosphomotifs remain phosphorylated during the transition. Indeed, the position (upstream or downstream), the nature (basic or acidic) and the length of specific sequences around the phosphoacceptors affects the phosphorylation of these sites during meiosis I exit. We found that phosphomotif behavior was the opposite when comparing mitotic and meiosis I exit. Importantly, when we inhibited Cdk activity in metaphase I, all kinase consensus motif phosphorylation waves were reversed back to a mitotic exit-like configuration. As in mitosis, Cdk is likely to provide priming phosphorylation on substrates, allowing other kinases to dock and phosphorylate nearby phosphosites, or to directly regulate the activity of kinases

and phosphatases. Overall, at both mitotic and meiosis I exit, Cdk activity levels determine which phosphomotifs are regulated, but only at mitotic exit the cell cycle is able to order the timing of substrate dephosphorylation.

We speculate that the qualitative model, based on specific kinase and phosphatase affinities, is the dominant model during the MI-MII transition. We have shown that some Cdk or Plk kinase consensus motifs are differentially regulated by the artificial drop of Cdk activity depending on the amino acid composition. Also, we have shown that many phosphosites (not classical Cdk sites) are still dephosphorylated during meiosis I exit, almost as much as during mitotic exit. This difference could be due to meiosis-specific kinase regulators that would modulate the catalytic site of mitotic kinases to recognize new kinase consensus motifs specifically in meiosis. For example, it has recently been shown by a comparable phosphoproteome analysis that the meiosis-specific protein Spo13 alters the substrate preference of Plk kinase when they interact in meiosis I (preprint: Koch et al, 2023).We suspect that other mitotic kinases and phosphatases are adapted through meiosis-specific regulation or interaction partners leading to meiosis-specific consensus phosphorylation. In addition, meiosis-specific kinases or phosphatases may act on the same substrates as are modified in mitosis, but at different sites. However, different combinations of phosphorylations can be redundant and result in the same output (Conti et al, 2023). Interestingly, it has been shown in budding yeast that Ime2 (a meiosis-specific kinase) phosphorylates the same substrates as Cdk, but at distinct sites. Crucially, these sites are much more resistant to the Cdc14 phosphatase than the Cdk sites

(Holt et al, 2007). Altogether, there are multiple strategies used to differently regulate meiosis I and mitotic exit.

In support of the qualitative model, it is known that cyclins bind their substrates through different docking motifs conferring substrate specificity. It has recently been shown that Clb1 targets LxF motifs, whereas Clb3 shows a preference for PxF motifs. These motifs have to be present at a specific distance from the phosphorylation site to control phosphorylation (Ord and Loog, 2019; Ord et al, 2019). Successive waves of cyclin expression and degradation regulate the mitotic cell cycle, but the expression patterns of cyclins are more complex in meiosis. Clb1 and Clb3 are specifically active in meiosis I and II, respectively (Carlile and Amon, 2008), and the specific organization of cyclin expression profiles in meiosis may be essential to target a specific category of substrates with high affinity. Consistent with this hypothesis, cyclins are in part interchangeable during the budding yeast mitotic cell cycle but have much less plasticity in the meiotic cell cycle (Carlile and Amon, 2008; Pirincci Ercan et al, 2021). Future work will show in how far cyclin specificity is key for meiotic progression, and if the mechanism discovered here for meiosis I to meiosis II transition in budding yeast applies to other organisms.

## Methods

### Reagents and tools

See Table 1.

**Table 1. Reagents and tools.**

| Reagent/Resource | Reference or Source | Identifier or Catalog Number |
|---|---|---|
| **Experimental Models** | | |
| All Saccharomyces cerevisiae SK1 strains | | |
| MATa, ho::LYS2, lys2, leu2::hisG, his3::hisG ndt80::pGAL-NDT80::TRP1, ura3::pGPD1-GAL4(848).ER::URA3 MATα, ho::LYS2, lys2, leu2::hisG, his3::hisG ndt80::pGAL-NDT80::TRP1, ura3::pGPD1-GAL4(848).ER::URA3 | Gift from Folkert van Werven | N/A |
| MATa, ho::LYS2, lys2, ura3, leu2::hisG, his3::hisG, trp1::hisG GAL-NDT80::TRP1 ura3::pGPD1-GAL4(848).ER::URA3 cdc28-as1 (F88G) MATalpha, ho::LYS2, lys2, ura3, leu2::hisG, his3::hisG, trp1::hisG GAL-NDT80::TRP1 ura3::pGPD1-GAL4(848).ER::URA3 cdc28-as1 (F88G)" | Gift from Folkert van Werven | N/A |
| **Antibodies** | | |
| Rat monoclonal anti- a-tubulin (YOL1/3) | Bio-Rad | Cat# MCA78G |
| Anti-rabbit IgG (HRP-conjugated) | GE Healthcare | Cat# NA934 |
| Phospho-MAPK/CDK Substrates (PXSP) | Ozyme | Cat# 2325 S |
| Phospho-CDK substrate [pTPXK] | Ozyme | Cat# 14371 S |
| Anti-alpha Tubulin antibody | abcam | Cat# ab4704 |
| α-rat CY3 | Interchim | Cat# 712-585-150 |
| **Chemicals, Enzymes, and other reagents** | | |
| Formaldehyde solution | Sigma-Aldrich | Cat# 252549 |
| cOmplete EDTA-Free Protease Inhibitor Cocktail | Sigma-Aldrich | Cat# 04693132001 |
| RNase A | Sigma-Aldrich | Cat# 10109169001 |
| Protein Assay Dye | Bio-Rad | Cat# 5000006 |

**Table 1.** (continued)

| Reagent/Resource | Reference or Source | Identifier or Catalog Number |
|---|---|---|
| Propidium iodide solution | Sigma-Aldrich | Cat# P4864 |
| β-Estradiol | Sigma-Aldrich | Cat# E2758 |
| 3-MB-PP1 | Sigma-Aldrich | Cat# 529582 |
| **Software** | | |
| FlowJo v10.1 | FlowJo | https://www.flowjo.com |
| Prism v7.0c | GraphPad | https://www.graphpad.com/scientific-software/prism/ |
| ImageJ v1.52c | ImageJ | https://imagej.nih.gov/ij/ |
| Perseus v1.4.0.2 | Perseus | https://maxquant.net/perseus/ |
| MaxQuant v1.6.17.0 | MaxQuant | https://maxquant.org |
| WebLogo 3 | WebLogo | https://weblogo.threeplusone.com/create.cgi |
| IceLogo | IceLogo | https://iomics.ugent.be/icelogoserver/ |
| Gene ontology | Gene ontology | http://www.geneontology.org |
| **Other** | | |
| ECL Prime Western Blotting Detection Reagent | GE Healthcare | Cat# RPN2232 |
| TMT 10plex Isobaric Label Reagent Set 1 ×0.8 mg | ThermoFisher | Cat# 90110 |
| TMT11-131C Label Reagent | ThermoFisher | Cat# A37724 |
| High-Select Fe-NTA Phosphopeptide Enrichment kit | ThermoFisher | Cat# A32992 |
| Sep-Pak C18 Cartridge | Ugap | Cat# 1568946 |
| Pierce Graphite Spin Column | ThermoFisher | Cat# 88302 |

## Strains and culture

Budding yeast SK1 strain was used for meiotic cultures. On day 1, cells were grown at 30 °C in rich YP medium supplemented with 2% glucose (1% yeast extract, 2% peptone and adenine) to reach an $OD_{600} = 20$–30. The culture was then transferred into BYTA media (1% yeast extract, 2% tryptone, 1% potassium acetate and 50 mM potassium phthalate). To sporulation, cells were resuspended in SPO media pH = 7 (0.3% potassium acetate) with an $OD_{600} = 3$–4. Cell synchronization was achieved using the NDT80 prophase I arrest and release system (Carlile and Amon, 2008). Release was performed using β-estradiol (5 mM stock in ethanol) at 5 h or 5h30 after sporulation. In strain carrying the analogue-sensitive Cdk allele, Cdk activity was inhibited 60 min after β-estradiol induction by adding 10 μM of the ATP analogue inhibitor 3-MB-PP1.

## Fluorescence-activated cell sorting analysis of DNA content

Cell synchrony and cell cycle progression in all experiments were monitored by analyzing DNA content using fluorescence-activated cell sorting (FACS). Cells from 1 ml culture ($OD_{600} = 3$–4) were collected by centrifugation and fixed in 70% ethanol at 4 °C overnight. Cells were resuspended in 1 ml of RNase buffer (50 mM Tris/HCl pH 7.5 with 0.1 mg/ml RNase A) and incubated at 37 °C for 2–6 h before being resuspended in 0.4 ml of FACS buffer (200 mM Tris/HCl pH 7.5, 211 mM NaCl, 78 mM MgCl$_2$) containing 50 μg/ml propidium iodide. After sonication, cell suspensions were diluted in 0.4 ml of 50 mM Tris/HCl pH 7.5. The samples were processed on the MACSQuant Flow Cytometer Analyzer. The obtained datafiles were then analyzed using the MACSQuantify and FlowJo softwares (FlowJo LLC).

## Immunofluorescence

Immunofluorescence was performed on formaldehyde-fixed cells. Cells from 1 ml culture ($OD_{600} = 3$–4) were collected by centrifugation and fixed overnight at 4 °C in 1 ml of ice-cold fixation buffer (100 mM potassium phosphate pH 6.4, 0.5 mM MgCl$_2$, 3.7% formaldehyde). After centrifugation, cells were washed in 1 ml of fixation buffer without formaldehyde and then in 1 ml spheroplasting buffer (100 mM potassium phosphate pH 7.4, 0.5 mM MgCl$_2$, 1.2 M sorbitol). Cells were spheroplasted in 0.2 ml spheroplasting buffer supplemented with 28 mM β-mercaptoethanol and 25 U/μl lytikase by incubation at 37 °C for 45 min. Cells were pelleted, washed in 0.5 ml spheroplasting buffer and finally resuspended in 0.2 ml of the same buffer. A 15-well slide was coated with 0.1% polylysine, and then 10 μl of cell suspension was applied. After 15 min in a moist chamber, drops were aspirated and slides were incubated 3 min in a −20 °C methanol bath and transferred for 10 s to a −20 °C acetone bath. After acetone evaporation, cells were covered with 10 μl of blocking buffer (1% BSA in PBS) for 20 min. Spindles were then stained using an α-tubulin antibody followed by an α-rat CY3 coupled antibody. Cells were counterstained with the DNA binding dye DAPI. To obtain fluorescent images, an inverted Zeiss Axiovert 200 M microscope with a 100X/1.4 NA oil objective coupled to an EMCD camera was used. Spindles < 2 μm in length were scored as short (metaphase), while spindles that were 2 μm or longer were classified as long (anaphase).

## Western blotting

5 ml culture ($OD_{600}$ = 3–4) was resuspended in 1 ml of 20% trichloroacetic acid and kept overnight at 4 °C before being washed in 1 ml of 1 M Tris-Base. Pellets were resuspended in 100 µL of 4X Laemmli buffer with dithiothreitol (DTT), boiled for 5 min and processed using FastPrep-24 2 × 45s. Bio-Rad protein assay was used to evaluate protein concentration and 15 µg of protein was separated by SDS-polyacrylamide gel electrophoresis before being transferred to a nitrocellulose membrane. To assess Cdk phosphorylation, a Phospho-MAPK/CDK Substrates (PxpSP, pSPx(K/R)) 2325S rabbit antibody at 1:500 and CDK pTPxKR rabbit at 1:500 were used. Tubulin (55KDa) rabbit antibody at 1:10000 was also used. Appropriate secondary antibody coupled to HRP (1:5000) was used for chemilumiescience/ECL detection of the signal.

## Sample preparation for Mass Spectrometry

18 ml of culture ($OD_{600}$ = 3–4) was resuspended in 5 ml of 20% trichloroacetic acid. Pellets were washed two times in 10–20 ml acetone, and resuspended in 250 µl of beating buffer (8 M urea, 50 mM Hepes pH7.5–8, 5 mM EDTA, 1 mM DTT, 50 mM sodium fluoride (NaF), 1 mM sodium vanadate, complete protease inhibitor mix), and processed using FastPrep-24 2 × 45s. BCA protein assay was used to evaluate protein concentration. 100 µg of each protein sample was reduced with 10 mM DTT final for 30 min at 37 °C and alkylated with 20 mM final iodoacetamide for 30 min at room temperature in the dark. Samples were diluted below 1 M urea with 50 mM HEPES pH 8.5 prior to trypsin digestion overnight at 37 °C (enzyme/substrate ratio of 1/50). The tryptic peptides were desalted on Pierce Graphite Spin columns and the eluates dried by speedvacuum at 35 °C. The desalted peptides were dissolved in 50 mM TEAB (pH 8.5) and then mixed with TMT10plex (Figs. 1–4 and EV1–EV3) or TMT11plex (Figs. 5, 6 and EV4, EV5) Isobaric Label Reagent Set, dissolved in 100% acetonitrile for 1 h. The labeling reaction was stopped by adding 5% hydroxylamine for 15 min. Label efficiency was checked by a 45 min data dependent acquisition run. After confirmation of a labeling efficiency >99%, the 10 or 11 samples were combined, and peptides were desalted using a C18 SepPack column according to the manufacturer's protocol. Phosphopeptide enrichment was performed using the Fe-NTA phosphopeptide enrichment kit. The non-phosphorylated peptides, the flow-through (FT) fractions, were kept for LC-MS/MS analysis. Phosphopeptides were eluted, dried in speed-vacuum and resuspended in FA 0.1%. 1/3 were loaded and desalted onto evotips provided by Evosep according to manufacturer's procedure. Phosphopeptides were analyzed in triplicate on an Orbitrap Fusion coupled to an Evosep One system operating with the 30SPD method developed by the manufacturer (Figs. 1–4 and EV1–EV3) and an Orbitrap Fusion (Figs. 5, 6 and EV4, EV5) mass spectrometer coupled to an UltiMate 3000 HPLC system for on line liquid chromatographic separation. Each run consisted of a three-hour gradient elution from a 75 µm × 50 cm C18 column. See also (Jones et al, 2020) for a very detailed description of the method.

## Quantification and statistical analysis

### *Mass Spectrometry data analysis of meiosis I exit datasets*

MaxQuant (version 1.6.17) was used for all data processing. The data was searched against a UniProt extracted *Saccharomyces*

*cerevisiae* proteome FASTA file (downloaded on April 2022 as FASTA file), amended to include common contaminants. Default MaxQuant parameters were used with the following settings: fixed modifications: carbamidomethyl (C), variable modifications: oxidation (M), acetyl (protein N-term), phosphorylation (STY), and deamidation (NQ); enzyme: Trypsin/P; max. missed cleavage: 2; first search peptide tolerance: 20 ppm; main search peptide tolerance: 4.5 ppm; second peptide search was enabled. All other settings were set to default. Results were filtered by 1% FDR on PSM and protein level. MaxQuant output files were imported into Perseus (version 1.6.2.3). All reporter intensities were $log_2$ transformed and only phosphosites that were quantified in all channels, 10 (Figs. 1–4 and EV1–EV3) or 11 (Figs. 5, 6 and EV4, EV5) were retained. Reporter intensities were normalized by subtracting the median value of each phosphosite across the time course. Smoothing was then performed by replacing each value by the mean of the two adjacent time points. For the smoothing of time points 50 and 140 min in Figs. 1–4 and EV1–EV3, only the 60 and 130 min values, respectively, were used. In Figs. 5, 6 and EV4, EV5, no smoothing was performed for the time point 0 min to have a common value in control and 3-MB-PP1 condition (Dataset EV1). For time point 25 min only time point 20 min was used. See the material and methods section of (Touati et al, 2018) for the mitotic exit dataset MS analysis.

### *Analysis of phosphosite abundance change*

The mitotic dataset was (Touati et al, 2018) reanalyzed using the same parameters for the mitotic exit and the meiosis I exit dataset (Figs. 1–4 and EV1–EV3). Phosphosite intensities were normalized to 1 at time point 50 min for the meiosis I exit dataset (time when the majority of cells are in metaphase I) and at time point 0 min for the mitotic exit dataset (time of metaphase release). All the analyses (except the ones mentioned in the figure legend) are in linear scale. A 1.5-fold decrease in phosphorylation abundance, or a 1.5-fold increase, over at least two consecutive time points was required to meet classification as "dephosphorylated" or "phosphorylated", respectively. Phosphosites that showed a less than 1.5-fold change were considered stable. For 3-MB-PP1 experiments (Figs. 5, 6, EV4, EV5), the ratio of 3-MB-PP1 data and control data was performed at each time point. A 2-fold decrease in phosphorylation abundance, or a 2-fold increase, over at least two consecutive time points was required to meet classification as "decrease" or "increase", respectively. Phosphosites that showed a less than 2-fold change were considered stable. For figures where more than one 3-MB-PP1 repeat is presented, datasets were normalized by subtracting the median of the log2 transformed values to obtain the same range of phosphorylation between the 3 datasets (Dataset EV1). In these conditions, a 1.5 cut off was used.

## IceLogo sequence analysis

Amino acid distributions surrounding phosphosites was analyzed using IceLogo https://iomics.ugent.be/icelogoserver/ (Colaert et al, 2009). Phosphoresidues are placed at the central position within the sequence logo. A *p*-value between 0.05 and 0.001 was applied as the detection threshold when using IceLogo. Percentage difference reflects the frequency of an amino acid in the indicated category, at a given position, compared to its frequency in the whole or the part of the dataset, from where the category had been extracted.

## Gene ontology

Gene ontology (GO) cellular component annotation was performed on http://www.geneontology.org (Ashburner et al, 2000; Gene Ontology et al, 2023). *P* values were analyzed via the PANTHER Classification System on http://pantherdb.org/ and −log10 *p* values are shown for each category.

# Data availability

The full mass spectrometry proteomics data obtained in this study have been deposited with the ProteomeXchange Consortium via the PRIDE partner repository with the dataset identifier PXD043862.

# Peer review information

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

## Acknowledgements

We thank L. Koch and A. Marston for discussion of unpublished results prior to submission, and Frank Uhlmann for comments on the manuscript. We would like to thank Folkert van Werven for Saccharomyces cerevisiae strains, Emilie-Fleur Gautier, Cédric Broussard and Marjorie Leduc, from Plateforme Proteom'IC 3P5, Université Paris Cité, Institut Cochin, Paris, France for performing sample acquisitions, and team members for discussion and critical reading of the manuscript. SAT received funding for this work by the Agence Nationale de la Recherche (ANR-21-CE13-0026), the Fondation de la Recherche contre le Cancer (ARCPJA12020060002043), the CNRS and Tremplin Sorbonne Université 2020.

## Author contributions

**Dunja Celebic**: Formal analysis; Investigation; Writing—review and editing. **Irem Polat**: Formal analysis; Investigation; Writing—review and editing. **Veronique Legros**: Data curation; Formal analysis; Investigation; Methodology. **Guillaume Chevreux**: Data curation; Supervision; Project administration. **Katja Wassmann**: Supervision; Project administration; Writing—review and editing. **Sandra A Touati**: Conceptualization; Data curation; Formal analysis; Supervision; Funding acquisition; Validation; Investigation; Visualization; Methodology; Writing—original draft; Project administration; Writing—review and editing.

## Disclosure and competing interests statement

The authors declare no competing interests.

# Expanded View Figures

**Figure EV1.  The phosphorylation landscape of the MI-MII transition in budding yeast.**

(**A**) Protein extracts were prepared at the indicated times and processed for Western blotting against the indicated proteins to monitor synchrony of the pre-meiotic phases. One representative experiment from three independent experiments is shown. (**B**) Percentage of phosphosites in each category of each repeat experiment. (**C**) Overlap between the proteins quantified in the proteome dataset (green) and the phosphoproteome (gray). (**D**) Fractions of proteins identified in the proteome dataset (non-enriched). Protein abundance changes through meiosis I exit are represented in gray (increase), in yellow (decrease) or in black (remain stable). (**E**) Overlap of phosphosites between the core experiment (repeat 1) and the main repeat (repeat 2). Phosphosites are categorized depending on the phosphorylation abundance changes during meiosis I exit - dephosphorylated in red, phosphorylated in blue and remained stable in black. (**F**) Profile plot of the 10 phosphosites dephosphorylated the earliest in the dataset. Timing of anaphase I is indicated. Note that many proteins play a role in the regulation of the synaptonemal complex. (**G**) Profile plot of the 10 phosphosites dephosphorylated the latest in the dataset. Timing of anaphase I is indicated. Note that many proteins are specific and essential for meiotic regulation. (**H**) Profile plot of the 10 phosphosites phosphorylated the earliest in the dataset. Timing of anaphase I is indicated. Note that many proteins are dephosphorylated after being phosphorylated early. Phosphosite intensities are shown on a log scale. (**I**) Profile plot of the 10 phosphosites phosphorylated the latest in the dataset. Timing of anaphase I is indicated. Phosphosite intensities are shown on a log scale.

▶

# A

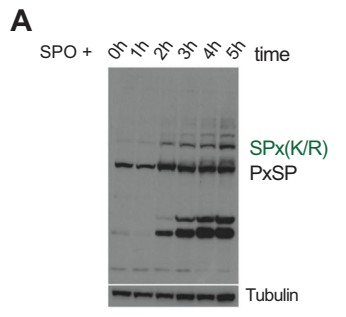

SPO +    0h  1h  2h  3h  4h  5h   time

SPx(K/R)
PxSP

Tubulin

# B

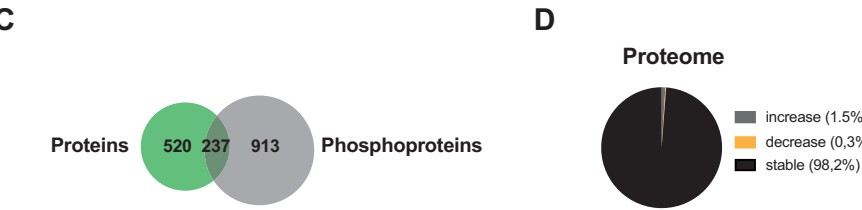

**Repeat 1**

|  | Proteome | Phosphoproteome |
|---|---|---|
| Identified | 757 | 4095 |
| Quantified | 757 | 2501 |
| Stable | 744 | 1730 |
| Increase | 11 | 423 |
| Decrease | 2 | 348 |

**Repeat 2**

|  | Proteome | Phosphoproteome |
|---|---|---|
| Identified | 828 | 2685 |
| Quantified | 828 | 1689 |
| Stable | 761 | 965 |
| Increase | 40 | 373 |
| Decrease | 27 | 361 |

# C

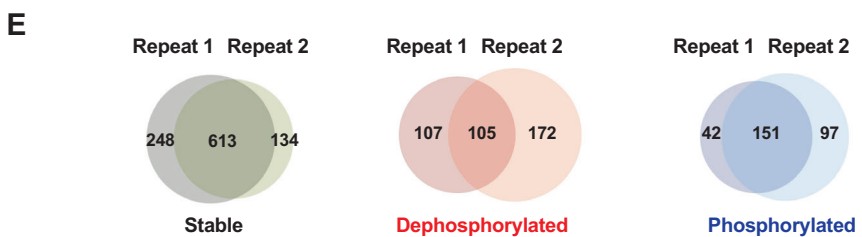

Proteins  520  237  913  Phosphoproteins

# D

**Proteome**

- increase (1.5%)
- decrease (0,3%)
- stable (98,2%)

# E

Repeat 1  Repeat 2

248  613  134

**Stable**

Repeat 1  Repeat 2

107  105  172

**Dephosphorylated**

Repeat 1  Repeat 2

42  151  97

**Phosphorylated**

# F

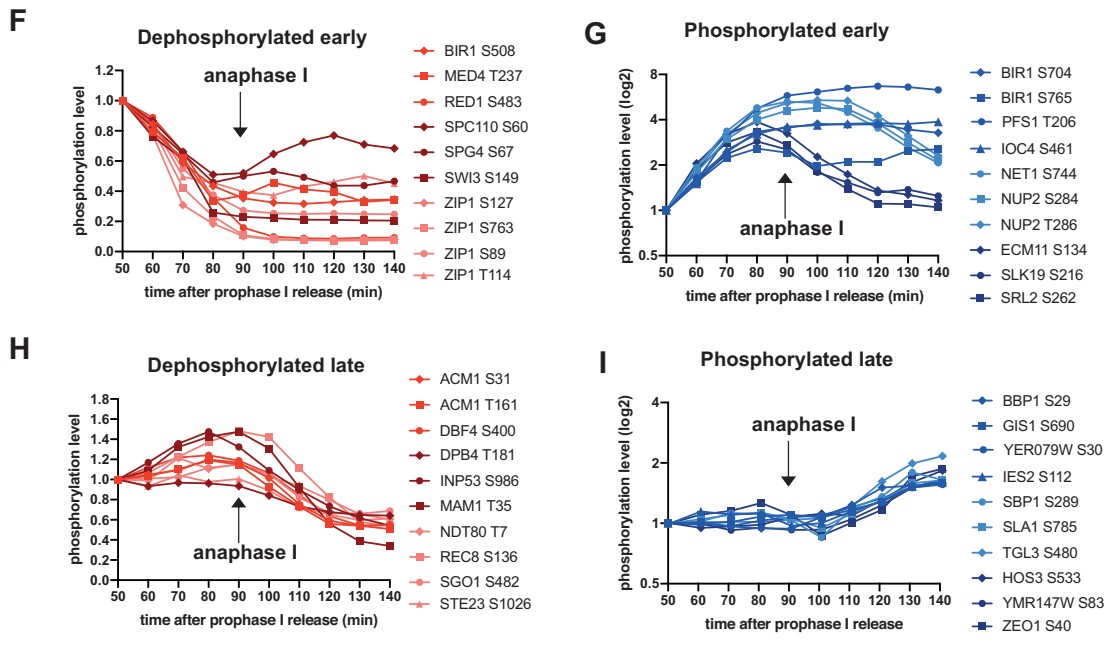

**Dephosphorylated early**

anaphase I

- BIR1 S508
- MED4 T237
- RED1 S483
- SPC110 S60
- SPG4 S67
- SWI3 S149
- ZIP1 S127
- ZIP1 S763
- ZIP1 S89
- ZIP1 T114

# G

**Phosphorylated early**

anaphase I

- BIR1 S704
- BIR1 S765
- PFS1 T206
- IOC4 S461
- NET1 S744
- NUP2 S284
- NUP2 T286
- ECM11 S134
- SLK19 S216
- SRL2 S262

# H

**Dephosphorylated late**

anaphase I

- ACM1 S31
- ACM1 T161
- DBF4 S400
- DPB4 T181
- INP53 S986
- MAM1 T35
- NDT80 T7
- REC8 S136
- SGO1 S482
- STE23 S1026

# I

**Phosphorylated late**

anaphase I

- BBP1 S29
- GIS1 S690
- YER079W S30
- IES2 S112
- SBP1 S289
- SLA1 S785
- TGL3 S480
- HOS3 S533
- YMR147W S83
- ZEO1 S40

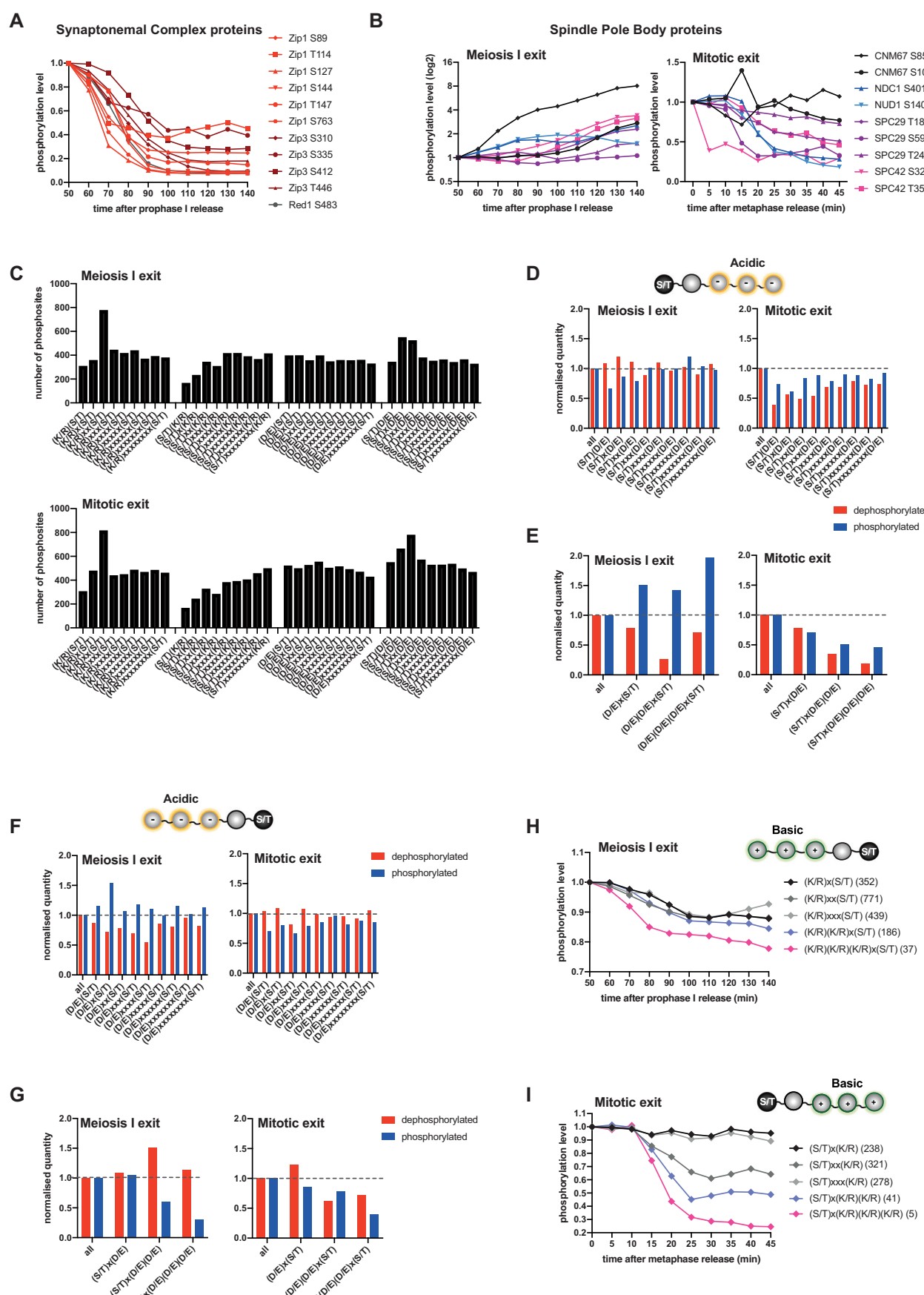

◀

**Figure EV2. Phosphorylation dynamics of mitotic kinase consensus motifs are reversed at meiosis I exit.**

(A) Profile plot examples of phosphosites on proteins related to the synaptonemal complex extracted from the meiosis I exit dataset. (B) Profile plot examples of phosphosites on proteins related to the spindle pole body extracted from the meiosis I exit dataset (left) and from the mitotic exit dataset (right). Phosphosite intensities are shown on a log scale on the left and on a linear scale on the right. (C) Number of phosphosites categorized by amino acid motif identity in the meiosis I exit dataset (above) and the mitotic exit dataset (below). (D–G) Percentage of dephosphorylated and phosphorylated sites categorized by amino acid motif identity after normalization to the total amount of sites. Phosphosites with distinct acidic downstream amino acids are presented in (D); with single, double, or triple acidic downstream amino acids in (E); with distinct acidic upstream amino acids in (F); and with single, double, or triple acidic upstream amino acids in (G). (H,I) Median intensity profile of the phosphosites categorized by amino acid motif identity through meiosis I exit. Phosphosites with basic upstream amino acids are presented in (H) and with basic downstream amino acids are presented in (I).

**A**

|  | | Meiosis I exit | | |
|---|---|---|---|---|
| | | Phosphorylated | Stable | Dephosphorylated |
| **Mitotic exit** | **Phosphorylated** | 4 | 39 | 13 |
| | **Stable** | 43 | 401 | 93 |
| | **Dephosphorylated** | 39 (3) | 86 (2) | 9 (1) |

**B**

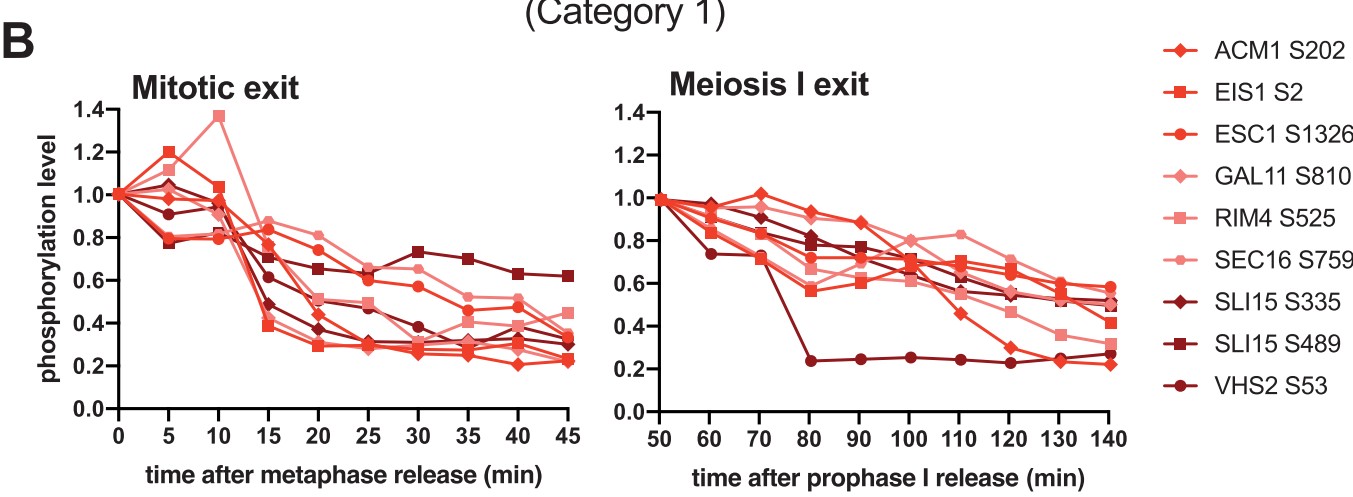

**C**

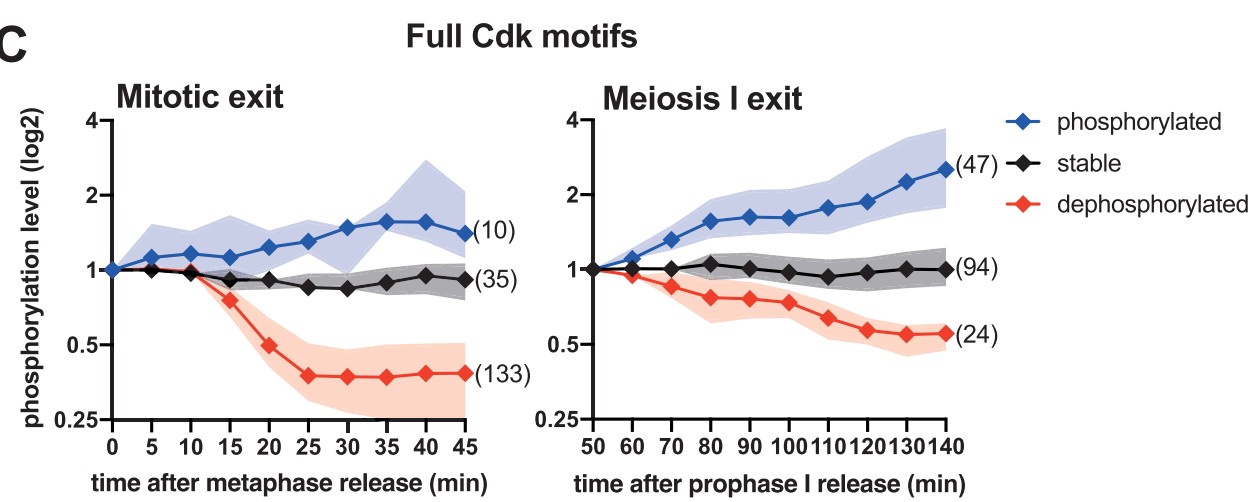

**Figure EV3. Full Cdk sites dephosphorylated during mitotic exit are mostly stably phosphorylated during meiosis I exit.**

(A) Table showing the number of phosphosites that are in common between the mitotic exit dataset and the meiosis I exit dataset in each category. (B) Profile plot of the 9 phosphosites dephosphorylated in both the mitotic exit and meiosis I exit dataset. (C) Normalized median intensity profiles and interquartile range of the full Cdk sites (S/T)Px(K/R) that undergo a 1.5-fold decrease (red) or a 1.5-fold increase (blue) in phosphorylation abundance through mitotic exit (left) and meiosis I exit (right). Normalized median intensity profiles and interquartile range of the phosphosites remaining stable are in black. Phosphosite intensities are shown on a log scale.

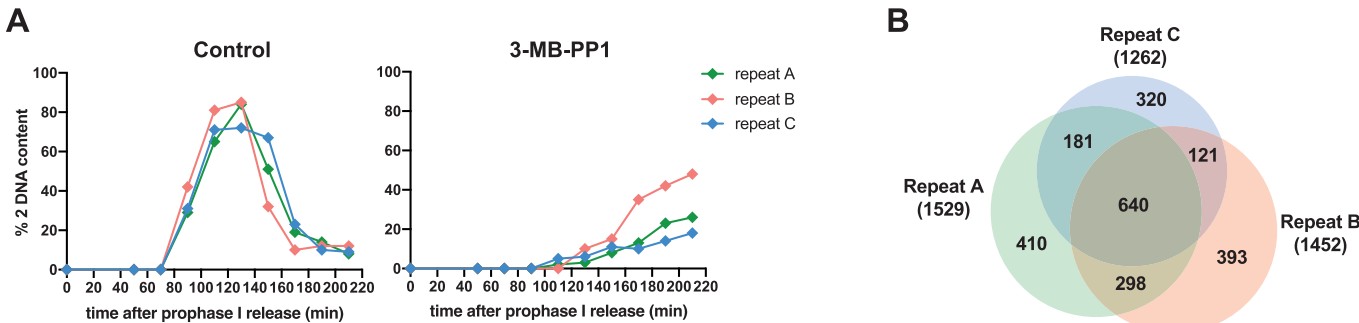

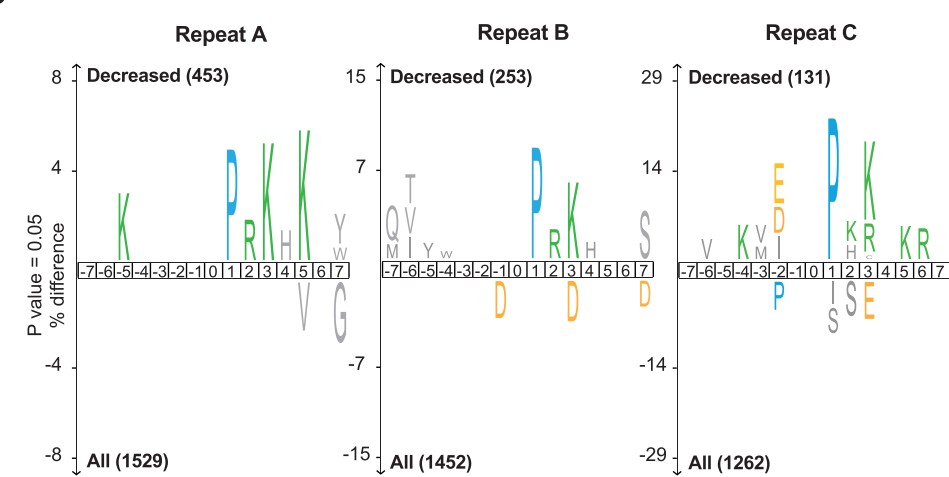

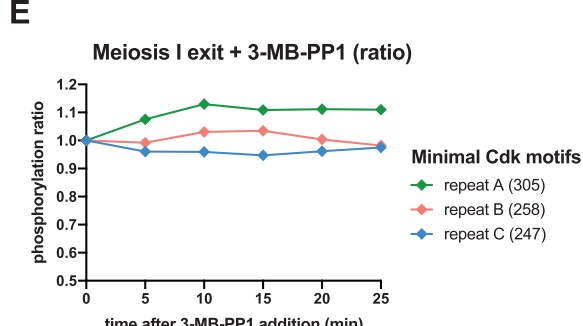

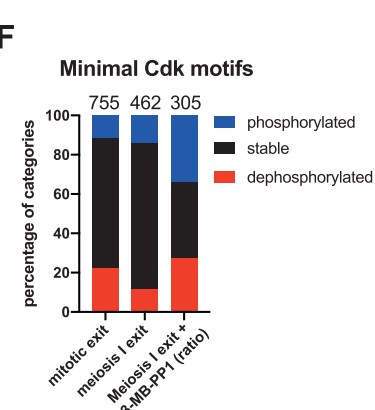

◀  **Figure EV4.  Phosphorylation landscape at meiosis I exit after complete drop in Cdk activity.**

(A) One hundred cells were scored for 2 DNA content at each timepoint (0 to 220 min after β-estradiol addition) in absence or presence of 3-MB-PP1 to determine the cell cycle phases. The timing for the three repeats is presented. (B) Overlap of phosphosites between the three repeats. (C) Table showing the percentage of phosphosites decreased or increased in each repeat. (D) IceLogo analysis in the three repeats highlighting phosphomotifs where phosphorylation abundance decreases after addition of 3MB-PP1. The phosphorylated residue is at position 0. Larger letter size indicates increased enrichment. Percentage of difference is used as scoring method. The threshold for enrichment detection was $p = 0.05$. (E) Median intensity profile of sites phosphorylated on the minimal Cdk motifs in the 3 repeats. Ratio values are plotted. (F) Percentage of the minimal Cdk sites (S/T)P in each category. Of note, this category excludes full Cdk sites (S/T)Px(K/R).

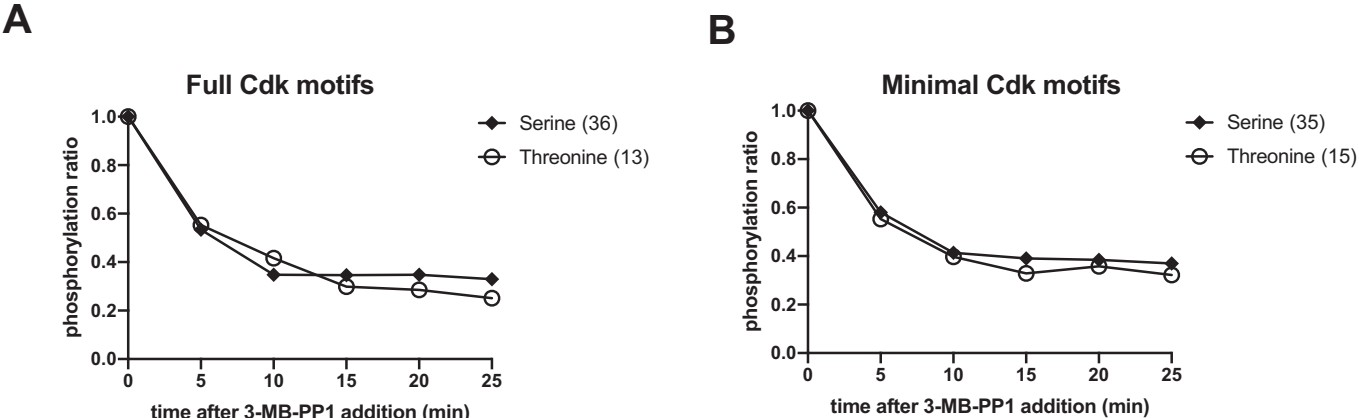

**Figure EV5.  Certain phosphorylation patterns at meiosis I exit remain meiosis-specific despite Cdk inhibition.**

(A,B) Normalized median intensity profiles of the phosphosites that decrease and adhere to full Cdk motifs (**A**) or minimal Cdk motifs (**B**).

