## [Peer Review File · The EMBO Journal]

Qualitative rather than quantitative phosphoregulation shapes the end of meiosis I in budding yeast

Sandra Touati, Dunja Celebic, Irem Polat, Veronique Legros, Guillaume Chevreux, and Katja Wassmann
DOI: 10.15252/embj.2023115446

Corresponding author(s): Sandra Touati (sandra.touati@ijm.fr)

Review Timeline:

Submission Date:	28th Aug 23
Editorial Decision:	13th Oct 23
Revision Received:	24th Nov 23
Editorial Decision:	2nd Jan 24
Revision Received:	4th Jan 24
Accepted:	5th Jan 24

Editor: Hartmut Vodermaier

Transaction Report:

Dr. Sandra A Touati
CNRS - Institut Jacques Monod
15 rue Helene Brion
Paris 75013
France

13th Oct 2023

Re: EMBOJ-2023-115446

Qualitative rather than quantitative phosphoregulation shapes the end of meiosis I in budding yeast

Dear Sandra,

Thank you again for submitting your study on meiosis I phosphoregulation to The EMBO Journal. We have now finally received a complete set of comments from three expert referees, copied below for your information. As you will see, all reviewers acknowledge the potential importance and interest of the work. Nevertheless, they raise a number of specific concerns, many of them related to presentation and interpretation of the data, that would need to be addressed prior to publication. Given that the referees' queries and issues appear for the most part straightforward to address, we would be happy to consider a revised manuscript further for publication.

Since it is our policy to pursue only a single round of major revision and therefore important to fully answer to all comments at the time of resubmission, please do not hesitate to get back to me with a tentative response letter/revision plan, in case you would like to clarify/discuss certain points already during the early stages of the revision. I should add that we could also offer extension of the default three-months revision period if needed, with our 'scooping protection' (meaning that competing work appearing elsewhere in the meantime will not affect our considerations of your study) remaining of course valid also throughout this extension.

Detailed information on preparing, formatting and uploading a revised manuscript can be found below and in our Guide to Authors. Thank you again for the opportunity to consider this work for The EMBO Journal, and I look forward to your revision in due time.

With kind regards,

Hartmut

9) Digital image enhancement is acceptable practice, as long as it accurately represents the original data and conforms to community standards. If a figure has been subjected to significant electronic manipulation, this must be clearly noted in the figure legend and/or the 'Materials and Methods' section. The editors reserve the right to request original versions of figures and the original images that were used to assemble the figure. Finally, we generally encourage uploading of numerical as well as gel/blot image source data; for details see: embopress.org/page/journal/14602075/authorguide#sourcedata

At EMBO Press, we ask authors to provide source data for the main manuscript figures. Our source data coordinator will contact you to discuss which figure panels we would need source data for and will also provide you with helpful tips on how to upload and organize the files.

In the interest of ensuring the conceptual advance provided by the work, we recommend submitting a revision within 3 months (11th Jan 2024). Please discuss the revision progress ahead of this time with the editor if you require more time to complete the revisions. Use the link below to submit your revision:

Link Not Available

Referee #1:

In the current study, Sandra Touati and colleagues compare Cdk phosphomotifs at the exit from mitosis and exit from meiosis I in budding yeast using global time-resolved phosphoproteomics. They found that Cdk phosphomotifs remain mostly stably phosphorylated at the end of meiosis I, whereas a majority of Cdk-unrelated motifs are reset by dephosphorylation. The study revealed opposite patterns of kinase consensus motif phosphorylation when comparing mitotic and meiosis I exit. Inducing an artificial drop of Cdk at the metaphase of meiosis I lead to ordered substrate dephosphorylation, comparable to the one observed in mitosis.

The authors conclude that the phosphoregulation of substrates at the end of meiosis I is mainly qualitative rather than quantitatively ordered.

The study is technically well performed and provides important and unexpected results. I would recommend it for publishing after the authors answer the points raised below.

1) The authors write that it is currently unknown whether the MI-MII transition depends on cyclin specificity or on the Cdk thresholds. This aspect needs a little more background information. For example, what are the cyclin expression dynamics during MI and MII?

2) In Fig 2, the switch of charges in the consensus sequence, on both C- and N-terminal sides, is very interesting. The authors should try to offer a broader reason or meaning for such an observation in the discussion section.

- 3) This result is quite surprising: the minimal Cdk motifs were not decreased after 3-MB-PP1 treatment, which confirms that very few (S/T)P motifs are direct targets of Cdk (Figure S4E). In theory, CDK should phosphorylate plenty of suboptimal S/TP sites. Could it be due to a meiosis-specific phenomenon and dependent on specific phosphatases? Or perhaps, the CDK-targeted suboptimal S/TP sites are phosphorylated at low levels (like the non-conventional CDK sites) and are not that well detectable in MS, while the detected S/TP sites could be targets of some other proline-directed kinases?
- 4) The authors state that it has been shown in mammals that the Cks module alters the Cdk catalytic pocket upon its association with a G1 or S-phase cyclin allowing mammalian Cdk1 to phosphorylate more « exotic » sites, i.e. without proline in +1 (Al-Rawi et al, 2023; Suzuki et al., 2015). In fact, there is no structural evidence that Cks1 alters the catalytic pocket. The non-conventional sites become phosphorylated due to the higher effective concentration of these sites in the active site vicinity, achieved via docking of phosphorylated sites to Cks1 or cyclin docking pockets.
- 5) The authors suspect that mitotic kinases and phosphatases are adapted through meiosis-specific regulation or interaction partners leading to meiosis-specific consensus phosphorylation. Could they, please, explain this idea a bit further?
- 6) The authors could consider and discuss proline isomerization as a potential way to differentiate between phosphorylated S/T-P sites in different physiological processes of the cell. Cis and Trans isomers can be differentially recognized by phosphatases. Is there any evidence of altered isomerization ratios?

Referee #2:

This study by Celebic et al uses a mass spectrometry phosphoproteomic approach to compare the changes in phosphosite utilization between exit from mitosis and exit from meiosis I. This question is of importance as cells exiting mitosis proceed into the next G1 and S phase whereas cells exiting meiosis I proceed directly into a second round of chromosome segregation. The authors analyze phosphopeptides isolated from whole cells in 10 minute intervals spanning from exit from meiosis I prophase through meiosis II (ie across the time of meiosis I exit). For comparison, they use a similar analysis of mitotic exit published earlier by the same laboratory. Rather than examine specific sites, the authors use the behavior of different classes of phosphosites (eg. CDK consensus, Cdc5 consensus, and Ipl1 consensus) to analyze the data. While the numbers of total phosphosites and the fraction that go up or down after exit are similar, the behavior of the types of sites is completely reversed. In mitotic exit, CDK consensus sites, as a class, are dephosphorylated and Cdc5 sites increase and the reverse is true in meiosis I exit. Moreover, an unbiased analysis reveals opposing behaviors for phosphosite utilization in the two conditions based on the acidic or basic character upstream or downstream of the phosphoacceptor site. Though the kinases (or phosphatases) whose activity produces these patterns is not clear, it is nonetheless a striking difference between the two conditions. Canonical CDK phosphosite usage was found to remain high in meiosis I exit, unlike mitotic exit. To determine if maintenance of CDK activity is the basis for these global differences between mitotic and meiosis I exit, the authors examined the consequences of inhibiting Cdc28 at the time of meiosis I exit. Under these circumstances, the behavior of the different classes of phosphosites reverts to the mitotic pattern, indicating that persistent Cdc28 activity is responsible for driving the differences in global phosphorylation under the two conditions.

Other comments:

There are many different kinases with similar consensus site preferences. Therefore, a certain amount of circumspection is required when drawing conclusions about the role of specific kinases and the authors should be careful when assigning phosphorylations to a particular kinase. For instance, the data described on p. 15 showing that Nx(S/T) sites behave differently than (D/E)x(S/T) sites is quite interesting but the conclusion that this due to effects on the activity of Plk1 is a leap. An at least equally likely alternative that should be considered is that there is another kinase (perhaps sporulation-specific) that recognizes these Nx(S/T) sites and is insensitive to Cdk activity.

For the full Cdk motifs in each class (Figure 3C) it would be helpful to have a list of the specific sites in a supplementary table.

Minor comments:

In Figure 1 C, no interquartile range seems to be shown for the dephosphorylated class of sites.

Figure 2, legend. In the section describing panels E-G, the text refers to "C", "D", and "E" when it should be "E", "F", and "G", respectively.

Figure 4C, What is meant by the labels "high" and "low"?

Figure 4F What is meant by "PxSP"?

Referee #3:

Dephosphorylation of CDK substrates is essential for cells to exit from mitosis and enter the next cell cycle. Unlike mitotic cells, meiotic cells must complete cell division twice without proceeding through DNA replication. How cell cycle regulation is altered to allow cell division without intervening DNA replication is not well understood. Here, the authors use phosphoproteomics to investigate whether differences in dephosphorylation of CDK substrates may account for this difference. The authors found that while dephosphorylation of CDK substrates is prevalent during mitotic exit (from their previous work; Touati et al, 2018), phosphorylation of CDK sites was far more stable at the MI-MII transition (this study). Interestingly, when CDK activity was artificially reduced during MI, cells exhibited the phosphoregulatory signature that is characteristic of mitotic exit.

Overall, this study provides valuable insights into mechanisms of meiotic regulation that are likely to be conserved among eukaryotes. The findings are interesting, the experiments are well-thought out, and the data presented clearly. However, there are a few issues that should be addressed to strengthen this manuscript. In particular, there are several places where conclusions appear to be drawn from a single time course replicate and additional analyses or experiments are necessary to strengthen the conclusions.

Major concerns

- 1) In the first paragraph of the results it states, "two repeats were performed and the one displaying the larger number of phosphosites quantified was used to illustrate the results". In Sup Fig S1E it appears that there is good reproducibility between replicates for sites that are stable or gain phosphorylation, but the overlap of sites that are dephosphorylated are poor. For this reason, it is important to repeat the key analyses with the second dataset, to see if the patterns observed hold in both replicates. For example, in Fig 2 and Fig 4 how similar are the analyses of dephosphorylated/phosphorylated motifs between the replicates? For Fig 2A the authors conclude "dephosphorylation is only prominent during mitotic exit", yet the differences in percentages seem modest. What are the percentages for the second meiotic dataset?
- 2) Strong conclusions can't be drawn from the changes in individual phosphosites if the data are based on a single replicate. For figures where individual sites are graphed (1E-G, 3D, 4G-I, 5D-E, 6B, S1F-I, S3B), how many of these sites were quantified in both replicates of the meiotic time course? How do the results compare? This is especially important for Fig 4G-I where the authors draw conclusions about phosphatase activity based on the changes in a few specific sites, presumably in only one replicate of the time course. Additional evidence should be provided to support these conclusions, and/or the text should be changed to indicate that the proposed model is speculative.
- 3) Fig 2B shows the enriched GO terms for proteins whose phosphorylation changes in either meiosis or mitosis. How many of the proteins that drive the significant enrichments are detected in both the mitotic and meiotic datasets? Are the differential enrichments driven by differential expression of proteins, or differences in phosphoregulation?
- 4) For the *cdc28-as1* inhibition experiments presented in Fig 5B, the authors state they use a high concentration of the inhibitor 3MB-PP1 for complete inhibition. Can they comment on why they used 3MB-PP1 when most studies use 1NM-PP1 as an inhibitor? Also, the data presented does not support their conclusion that complete inhibition was achieved. The Western in Fig 5C shows some phosphorylations persist, although this could just mean that some sites are very stable. Ideally, complete inhibition would be demonstrated using an IP-kinase assay. Alternatively, the authors should show a phospho-TPx(K/R) Western blot alongside the phospho-SPx(K/R) blot to determine if the TP sites decrease similarly to SP sites, and the caveats of using a Western as a readout for kinase activity should be discussed.

Minor Concerns

- 5) In Fig 1C & 2C, median and interquartile range are plotted for different categories of phosphosites. The IQR should also be included in similar graphs later in the paper to show the variability in the data (4D, 5F, most of Fig 6).
- 6) I found the title and the discussion of the results in terms of a qualitative vs quantitative difference to be confusing. The evidence for qualitative regulation by CDK and phosphatases in budding yeast is strong, so it seemed unlikely that quantitative regulation would explain the difference between mitotic and meiotic exit. Although I believe their claims about qualitative regulation are correct, I don't think the discussion of qualitative vs quantitative adds much to this story and would consider removing it from the title and introduction. I don't feel strongly about this however, it's ultimately up to the authors.
- 7) The description of the NDT80 shut off system at the beginning of the results (and depicted in Fig 1A) is difficult to follow if the reader is unfamiliar with this system. A more complete description of how the system works in the text (for instance, what does *b-estradiol* do?) would be helpful. In addition, the labels for the graph in 1A (1DNA, 2DNA, 4DNA) should be defined.
- 8) The y-axes in Fig 2C should be labeled.

EMBOJ-2023-115446

Qualitative rather than quantitative phosphoregulation shapes the end of meiosis I in budding yeast

Referee #1:

In the current study, Sandra Touati and colleagues compare Cdk phosphomotifs at the exit from mitosis and exit from meiosis I in budding yeast using global time-resolved phosphoproteomics. They found that Cdk phosphomotifs remain mostly stably phosphorylated at the end of meiosis I, whereas a majority of Cdk-unrelated motifs are reset by dephosphorylation. The study revealed opposite patterns of kinase consensus motif phosphorylation when comparing mitotic and meiosis I exit. Inducing an artificial drop of Cdk at the metaphase of meiosis I lead to ordered substrate dephosphorylation, comparable to the one observed in mitosis.

The authors conclude that the phosphoregulation of substrates at the end of meiosis I is mainly qualitative rather than quantitatively ordered.

The study is technically well performed and provides important and unexpected results. I would recommend it for publishing after the authors answer the points raised below.

We thank the reviewer for highlighting the novelty of our findings and recommending publication.

1) The authors write that it is currently unknown whether the MI-MII transition depends on cyclin specificity or on the Cdk thresholds. This aspect needs a little more background information. For example, what are the cyclin expression dynamics during MI and MII?

We have followed the reviewer's comment and added some information about the meiotic cyclin activity in the introduction.

2) In Fig 2, the switch of charges in the consensus sequence, on both C- and N-terminal sides, is very interesting. The authors should try to offer a broader reason or meaning for such an observation in the discussion section.

We agree with this reviewer that the switch of charges is a very interesting observation. We have added a phrase in the discussion to emphasize this issue.

3) This result is quite surprising: the minimal Cdk motifs were not decreased after 3-MB-PP1 treatment, which confirms that very few (S/T)P motifs are direct targets of Cdk (Figure S4E). In theory, CDK should phosphorylate plenty of suboptimal S/TP sites. Could it be due to a meiosis-specific phenomenon and dependent on specific phosphatases? Or perhaps, the CDK-targeted suboptimal S/TP sites are

phosphorylated at low levels (like the non-conventional CDK sites) and are not that well detectable in MS, while the detected S/TP sites could be targets of some other proline-directed kinases?

We thank the reviewer for this comment and have noted that our previous statement was confusing. Figure EV4E shows the median profiles of minimal Cdk sites, and as the majority of these are not dephosphorylated, they look flat, suggesting that they are little affected by Cdc28 inhibition. During mitotic exit, when Cdk activity decreases, the vast majority of (S/T)P sites also remain phosphorylated (only 22% are dephosphorylated) (previous Figure EV3D and new Figure EV4F). In fact, a similar number (27% dephosphorylated) is observed in the context of meiosis I exit + 3MBPP1. To avoid any misunderstanding, we have now moved Figure EV3D to Figure EV4F and combined the information. Note that we have also observed that (S/T)P motifs are more likely to be dephosphorylated when they have a K or R in the +4-5 position (Figure 2F). For these reasons, we do not think it is a meiosis-specific phenomenon depending on other specific kinases or phosphatases, although this would have been a very interesting hypothesis.

4) The authors state that it has been shown in mammals that the Cks module alters the Cdk catalytic pocket upon its association with a G1 or S-phase cyclin allowing mammalian Cdk1 to phosphorylate more « exotic » sites, i.e. without proline in +1 (Al-Rawi et al, 2023; Suzuki et al., 2015). In fact, there is no structural evidence that Cks1 alters the catalytic pocket. The non-conventional sites become phosphorylated due to the higher effective concentration of these sites in the active site vicinity, achieved via docking of phosphorylated sites to Cks1 or cyclin docking pockets.

We agree with the reviewer that our statement was too strong. We have removed it as an example. In the revised version of the manuscript, we now include a recent example of changes in kinase specificity through interaction partners, just published in bioRxiv, which is more in line with our idea: In budding yeast meiosis, the fact that spo13 binds to Cdc5 (Plk) seems to increase the preference of Cdc5 for a specific type of phosphomotif that is different from the classical Cdc5 motif.

5) The authors suspect that mitotic kinases and phosphatases are adapted through meiosis-specific regulation or interaction partners leading to meiosis-specific consensus phosphorylation. Could they, please, explain this idea a bit further?

This idea is in line with point 4) and we have now better defined it in the discussion. To differentially modulate the meiotic and mitotic landscape, we hypothesised that 1) some meiosis-specific kinases (e.g. Ime2) with a meiosis-specific kinase consensus motif or 2) some meiosis-specific kinase regulators (e.g. Spo13) that would modulate the catalytic pocket of classical kinases (e.g. Cdc5) by creating new kinase consensus motifs, would allow phosphorylation of meiosis-specific phosphomotif different from those in mitosis (but potentially on the same substrates). The output for the substrate could then be different or similar (depending on phosphosite redundancy).

6) The authors could consider and discuss proline isomerization as a potential way to differentiate between phosphorylated S/T-P sites in different physiological processes of the cell. Cis and Trans isomers can be differentially recognized by phosphatases. Is there any evidence of altered isomerization ratios?

We agree with the reviewer's point, but unfortunately mass spectrometry is not yet a common tool for distinguishing cis/trans proline isomers. On the contrary, recent proof-of-concept studies on this topic using ion mobility suggest that mass spectrometry should be used very cautiously, as any thermal activation during ion transfer from the liquid to the gas phase or while in the gas phase may lead to proline isomer scrambling (JW Silzel et al., 2020 (doi: 10.1021/jasms.0c00242)). Furthermore, our study was not designed to determine the state of proline isomerisation and did not use an ion mobility spectrometer. Therefore, our data cannot be used to estimate the state of proline isomerisation in solution.

Referee#2:

This study by Celebic et al uses a mass spectrometry phosphoproteomic approach to compare the changes in phosphosite utilization between exit from mitosis and exit from meiosis I. This question is of importance as cells exiting mitosis proceed into the next G1 and S phase whereas cells exiting meiosis I proceed directly into a second round of chromosome segregation. The authors analyze phosphopeptides isolated from whole cells in 10 minute intervals spanning from exit from meiosis I prophase through meiosis II (ie across the time of meiosis I exit). For comparison, they use a similar analysis of mitotic exit published earlier by the same laboratory. Rather than examine specific sites, the authors use the behavior of different classes of phosphosites (eg. CDK consensus, Cdc5 consensus, and Ipl1 consensus) to analyze the data. While the numbers of total phosphosites and the fraction that go up or down after exit are similar, the behavior of the types of sites is completely reversed. In mitotic exit, CDK consensus sites, as a class, are dephosphorylated and Cdc5 sites increase and the reverse is true in meiosis I exit. Moreover, an unbiased analysis reveals opposing behaviors for phosphosite utilization in the two conditions based on the acidic or basic character upstream or downstream of the phosphoacceptor site. Though the kinases (or phosphatases) whose activity produces these patterns is not clear, it is nonetheless a striking difference between the two conditions. Canonical CDK phosphosite usage was found to remain high in meiosis I exit, unlike mitotic exit. To determine if maintenance of CDK activity is the basis for these global differences between mitotic and meiosis I exit, the authors examined the consequences of inhibiting Cdc28 at the time of meiosis I exit. Under these circumstances, the behavior of the different classes of phosphosites reverts to the mitotic pattern, indicating that persistent Cdc28 activity is responsible for driving the differences in global phosphorylation under the two conditions.

Other comments:

There are many different kinases with similar consensus site preferences. Therefore, a certain amount of circumspection is required when drawing conclusions about the role of specific kinases and the authors should be careful when assigning phosphorylations to a particular kinase. For instance, the data described on p. 15 showing that Nx(S/T) sites behave differently than (D/E)x(S/T) sites is quite interesting but the conclusion that this due to effects on the activity of Plk1 is a leap. An at least equally likely alternative that should be considered is that there is another kinase (perhaps sporulation-specific) that recognizes these Nx(S/T) sites and is insensitive to Cdk activity.

We agree with the reviewer that some kinase consensus motifs can be recognised by more than one kinase and that there is some overlap. For example, we highlight in our study the Rxx(S/T) motif can be recognised by Aurora, PKA and several MEN pathway kinases.

We also agree that we cannot rule out the possibility that a meiosis-specific kinase could target the Nx(S/T) motif and then keep this motif phosphorylated. To our knowledge, no meiosis-specific kinase has been identified that recognises this motif, but it is a very interesting possibility to explore. Alternatively, some meiosis-specific regulators could alter the phosphorylation preference of both a mitotic and meiotic kinase or phosphatase, leading to the maintenance of Nx(S/T) phosphorylation. We have added a sentence in the result section of the manuscript to address this comment of the reviewer.

For the full Cdk motifs in each class (Figure 3C) it would be helpful to have a list of the specific sites in a supplementary table.

We agree, and have now provided an Excel sheet (sheet 7) in the Dataset EV1 with the dynamics of full Cdk motifs during mitotic exit and meiosis I exit repeat 1 and 2.

Minor comments:

In Figure 1 C, no interquartile range seems to be shown for the dephosphorylated class of sites.

The interquartile range is shown, but there is only a small variation because the majority of dephosphorylated sites follow a similar profile.

Figure 2, legend. In the section describing panels E-G, the text refers to "C", "D", and "E" when it should be "E", "F", and "G", respectively.

We thank the reviewer for pointing this out, we have fixed this issue.

Figure 4C, What is meant by the labels "high" and "low"?

High and low should have been correlated to an exposure time in the figure legend, but this information was missing and we thank the reviewer for pointing this out. It has now been changed in the figure and figure legend.

Figure 4F What is meant by "PxSP"?

The phosphoantibody we use recognises not only the SPx(K/R) motif, but also the PxSP motif. This is mentioned in the Methods section and has been now added to the figure legend.

Referee#3:

Dephosphorylation of CDK substrates is essential for cells to exit from mitosis and enter the next cell cycle. Unlike mitotic cells, meiotic cells must complete cell division twice without proceeding through DNA replication. How cell cycle regulation is altered to allow cell division without intervening DNA replication is not well understood. Here, the authors use phosphoproteomics to investigate whether differences in dephosphorylation of CDK substrates may account for this difference. The authors found that while dephosphorylation of CDK substrates is prevalent during mitotic exit (from their previous work; Touati et al, 2018), phosphorylation of CDK sites was far more stable at the MI-MII transition (this study). Interestingly, when CDK activity was artificially reduced during MI, cells exhibited the phosphoregulatory signature that is characteristic of mitotic exit.

Overall, this study provides valuable insights into mechanisms of meiotic regulation that are likely to be conserved among eukaryotes. The findings are interesting, the experiments are well-thought out, and the data presented clearly. However, there are a few issues that should be addressed to strengthen this manuscript. In particular, there are several places where conclusions appear to be drawn from a single time course replicate and additional analyses or experiments are necessary to strengthen the conclusions.

We thank the reviewer for highlighting the value and interest of our findings.

Major concerns

1) In the first paragraph of the results it states, "two repeats were performed and the one displaying the larger number of phosphosites quantified was used to illustrate the results". In Sup Fig S1E it appears that there is good reproducibility between replicates for sites that are stable or gain phosphorylation, but the overlap of sites that are dephosphorylated are poor. For this reason, it is important to repeat the key analyses with the second dataset, to see if the patterns observed hold in both replicates. For example, in Fig 2 and Fig 4 how similar are the analyses of

dephosphorylated/phosphorylated motifs between the replicates? For Fig 2A the authors conclude "dephosphorylation is only prominent during mitotic exit", yet the differences in percentages seem modest. What are the percentages for the second meiotic dataset?

We fully understand the reviewer's concern and regret that the first version of the manuscript was not clear about the analysis of the repeats. We did not clearly mention that the conclusions we drew from the analysis of repeat 1 were also observed in repeat 2. The reviewer wonders about the percentage in categories of the second meiotic dataset. The number of phosphosites in each category is shown in Figure EV1B repeat 2. As the reviewer pointed out, the number of dephosphorylated and phosphorylated phosphosites is more similar in repeat 2 than in repeat 1. Nevertheless, full Cdk sites of repeat 2 have similar phosphorylation profiles to repeat 1 and the phosphomotifs are reversed between mitotic exit and meiosis I exit, as are the other kinase motifs. To clarify this point, we have now provided the Appendix Figure S1A-N where we have extracted all the information from repeat 2. We have also mentioned in the result part that the two repeats allow us to draw similar conclusions.

2) Strong conclusions can't be drawn from the changes in individual phosphosites if the data are based on a single replicate. For figures where individual sites are graphed (1E-G, 3D, 4G-I, 5D-E, 6B, S1F-I, S3B), how many of these sites were quantified in both replicates of the meiotic time course? How do the results compare? This is especially important for Fig 4G-I where the authors draw conclusions about phosphatase activity based on the changes in a few specific sites, presumably in only one replicate of the time course. Additional evidence should be provided to support these conclusions, and/or the text should be changed to indicate that the proposed model is speculative.

Independent mass spectrometry analyses do not allow the identification of exactly the same phosphosites in different data sets, this is inherent in the technology. In our analyses, we are rarely interested in individual phosphosites, we focus on global patterns and behaviour. However, there is an interest in the cell cycle community for having the global pattern illustrated by examples of substrates they can refer to. For this reason, we provide the examples shown in Figure 1E-G, 3D, 6B, S1F-I, S3B, which also illustrate the kind of kinetics we can get with time-resolved phosphoproteomic analysis. Figure 5D-E shows how the meiosis I exit + 3MBPP1 ratio is calculated, any other full Cdk motif could have been used; in fact, we could have even removed the name of the phosphosite.

Figure 5D-E shows the global behaviour of the full Cdk motif during mitotic exit versus meiosis I exit. Again, we want to show a global behaviour, and have chosen to highlight some phosphosites because of the reasons above. In the revised version of the manuscript, many of these phosphosites found in repeat 2 have been plotted in Appendix Figure S1J.

For the 4G-I phosphatase behavior, we agree with the reviewer that these sites are particularly important to draw conclusions. We observe similar behavior in repeat 2 as in repeat 1: Igo1 S64, which decreases over time, Cdc14 S429, which increases, and other Net1 sites known to be phosphorylated to allow Cdc14 activation, which follow a similar up and down profile in repeat 1 and 2. The plot of these sites can now be found in Appendix Figure S1N in the revised version of the manuscript. Moreover, the results found in these two repeats are consistent with what has already been suggested in the literature (Buonomo et al., 2003; Kamieniecki et al., 2005; Marston et al., 2003; Pablo-Hernando et al., 2007). Altogether, this allows us to speculate on phosphatases that orchestrate the MI-MII transition.

3) Fig 2B shows the enriched GO terms for proteins whose phosphorylation changes in either meiosis or mitosis. How many of the proteins that drive the significant enrichments are detected in both the mitotic and meiotic datasets? Are the differential enrichments driven by differential expression of proteins, or differences in phosphoregulation?

Due to the stability of the proteome at the MI-MII transition, we argue that the observed differences are due to phosphoregulation and not to proteome variation during the MI-MII transition.

In the GO term analysis, we do not compare the enrichment of mitotic and meiotic phosphoproteins, but the phosphorylation dynamics of the detected proteins, either phosphorylated or dephosphorylated. The figures for meiosis I exit and mitotic exit are independent of each other in terms of significance. The aim of the analysis was to see a difference in phosphorylation dynamics in a cellular component or biological process between mitotic exit and meiosis I exit. However, nothing striking emerged from this analysis, showing that phosphosites are more regulated at the level of phosphorylation motifs and substrates than at the level of protein groups. In this figure, we illustrate some results that make sense according to the mitotic or meiotic cell cycle, as the presence or absence of phosphoproteins involved in the synaptonemal complex at meiosis I exit and mitotic exit, respectively. Note that the same results are observed in repeat 2 Appendix Figure S1D. The take-home message is that synaptonemal complex proteins are regulated more by dephosphorylation than by phosphorylation during meiosis I exit. The proteins involved in the spindle pole body are more phosphorylated during meiosis I exit and dephosphorylated during mitotic exit.

4) For the *cdc28-as1* inhibition experiments presented in Fig 5B, the authors state they use a high concentration of the inhibitor 3MB-PP1 for complete inhibition. Can they comment on why they used 3MB-PP1 when most studies use 1NM-PP1 as an inhibitor? Also, the data presented does not support their conclusion that complete inhibition was achieved. The Western in Fig 5C shows some phosphorylations persist, although this could just mean that some sites are very stable. Ideally, complete

inhibition would be demonstrated using an IP-kinase assay. Alternatively, the authors should show a phospho-TPx(K/R) Western blot alongside the phospho-SPx(K/R) blot to determine if the TP sites decrease similarly to SP sites, and the caveats of using a Western as a readout for kinase activity should be discussed.

In agreement with the reviewer, both 3MB-PP1 and 1NM-PP1 drugs can be used to inhibit the Cdc28-as allele. In our preliminary experiments, we did not observe any difference when using one or the other. Since 3MB-PP1 is 5 times less expensive than 1NM-PP1, we used 3MB-PP1.

In Figure 5C, before performing the whole phosphoproteome analysis, we used phosphomotif antibodies on Western blots as a readout to check whether Cdk substrate phosphorylation changes after the addition of 3MB-PP1. We agree with the reviewer that this western blot does not allow us to assess Cdk activity directly and we have changed the text to avoid any confusion. As the antibody used in Figure 5C recognises two motifs, SPx(K/R) and PxSP, it is not surprising that only part of the bands observed are reduced after the addition of 3MB-PP1. As suggested by the reviewer, we also used TPx(K/R) phosphomimetic antibody in this experiment and we decided not to include the result in the article. In fact, after finishing the previous batch of TPx(K/R) antibody, we never got a signal as good as the one observed Figure 4C with our new batches. In addition, a band at around 100KDa is oversaturated (please see the blot below). Nevertheless, we can see that in presence of 3-MB-PP1 some bands no longer appear at the time of anaphase I, confirming that 3-MB-PP1 can also inhibits Cdk substrates with threonine residues. However, as we cannot repeat the western with the antibody batches currently available, we prefer to not include this data in the revised version of the manuscript.

Minor Concerns

5) In Fig 1C & 2C, median and interquartile range are plotted for different categories

of phosphosites. The IQR should also be included in similar graphs later in the paper to show the variability in the data (4D, 5F, most of Fig 6).

We have chosen to show the interquartile range only when it does not affect the visualisation of the data. For example, when the interquartile range is added to Figure 6D-L, the data become visually impossible to understand, because of the overlapping colours. Thus, we lose the point we want to make, in the figure. The interquartile range is a tool to help visualise the data, not the other way around, so we have decided to leave the figures as they are. However, in the revised version we provide all the raw data from the experimental dataset, allowing the reader to reanalyse the data in any way they wish. Note that we have added the interquartile range of Figure 6A because it does not interfere with the visualisation of the data and the Figure is thus consistent with Figure 2C.

6) I found the title and the discussion of the results in terms of a qualitative vs quantitative difference to be confusing. The evidence for qualitative regulation by CDK and phosphatases in budding yeast is strong, so it seemed unlikely that quantitative regulation would explain the difference between mitotic and meiotic exit. Although I believe their claims about qualitative regulation are correct, I don't think the discussion of qualitative vs quantitative adds much to this story and would consider removing it from the title and introduction. I don't feel strongly about this however, it's ultimately up to the authors.

We understand the author's point. However, the question of whether the cell cycle is mainly regulated by qualitative or quantitative regulation has been extensively studied in mitosis. *S. pombe*, for example, can only rely on quantitative regulation by Cdk, and in *S. cerevisiae*, cell cycle progression also seems to depend mainly on quantitative regulation by Cdk, with the exception of a few processes. In our opinion, the fact that this is clearly not the case in meiosis (at least in budding yeast) is the main message of our manuscript. Therefore, we prefer not to change the title and keep our focus on this key aspect.

7) The description of the NDT80 shut off system at the beginning of the results (and depicted in Fig 1A) is difficult to follow if the reader is unfamiliar with this system. A more complete description of how the system works in the text (for instance, what does b-estradiol do?) would be helpful. In addition, the labels for the graph in 1A (1DNA, 2DNA, 4DNA) should be defined.

We have followed the reviewer's advice and in the revised version of the manuscript we have explained in more detail the Ndt80 arrest and release system used. We have also added the definition of 1DNA, 2DNA, 4DNA to the figure legend.

8) The y-axes in Fig 2C should be labeled.

We thank the reviewer for pointing out this error, which has been corrected.

Dr. Sandra A Touati
CNRS - Institut Jacques Monod
15 rue Helene Brion
Paris 75013
France

2nd Jan 2024

Re: EMBOJ-2023-115446R
Qualitative rather than quantitative phosphoregulation shapes the end of meiosis I in budding yeast

Dear Sandra,

Thank you for submitting your revised manuscript to The EMBO Journal. Two of the original referees have now assessed it once again (see comments below), and I am happy to say that both of them are overall satisfied with your responses and revisions. Following a final round of minor revision to incorporate the remaining textual suggestions of referee 3, we should therefore be ready to accept the study for publication.

In addition, there are a number of editorial issues that need to be addressed at this stage:

- As we are switching from a free-text author contribution statement towards a more formal statement based on Contributor Role Taxonomy (CRediT) terms, please remove the present Author Contribution section and instead specify each author's contribution(s) directly in the Author Information page of our submission system during upload of the final manuscript. See <https://casrai.org/credit/> for more information.
- Please rename the Conflict of Interest section into "Disclosure and Competing Interests Statement", in accordance with our updated Guide to Authors (<https://www.embopress.org/competing-interests/>
- Please double-check all citations in the bibliography (which should be renamed to "References"), as several of them appear to be still incomplete (lacking volume/page/locator information). Also, when citing preprints, please make sure to adhere to the format specified in our Guide to Authors: The citation in the text should be: "(preprint: NAME1 et al, YEAR)"; in the reference list: "Author NAME1, Author NAME2, ... (YEAR) article title. bioRxiv (e.g.!) doi: XXX"
- Please ensure that data listed in the Data Availability section becomes publicly accessible at this point, latest upon formal acceptance.
- In the Appendix, please make sure to label the contents in full as "Appendix Figure S1" and "Appendix Figure S1 legend". Also, please remove its legend from the main text file, it should only be present in the Appendix file itself.
- Please upload the Source Data file folders, which are currently all combined in a single ZIP archive, as 6 individual ZIP archives, one per each main figure.
- Finally, during routine pre-acceptance checks, our data editors have raised the following queries regarding figures, data, and legends:
In the Figure Legends for main & EV figures, please indicate the statistical test used for data analysis in the legends of figures 2b, d; 3b; 5g; 6b, d; EV4d.

I am therefore returning the manuscript to you for a final round of minor revision, to allow you to make these adjustments and upload all modified files. Once we will have received them, we should be ready to swiftly proceed with formal acceptance and production of the manuscript.

With kind regards,

Hartmut

9) Digital image enhancement is acceptable practice, as long as it accurately represents the original data and conforms to community standards. If a figure has been subjected to significant electronic manipulation, this must be clearly noted in the figure legend and/or the 'Materials and Methods' section. The editors reserve the right to request original versions of figures and the original images that were used to assemble the figure. Finally, we generally encourage uploading of numerical as well as gel/blot image source data; for details see: embopress.org/page/journal/14602075/authorguide#sourcedata

At EMBO Press, we ask authors to provide source data for the main manuscript figures. Our source data coordinator will contact you to discuss which figure panels we would need source data for and will also provide you with helpful tips on how to upload and organize the files.

In the interest of ensuring the conceptual advance provided by the work, we recommend submitting a revision within 3 months (1st Apr 2024). Please discuss the revision progress ahead of this time with the editor if you require more time to complete the revisions. Use the link below to submit your revision:

Link Not Available

Referee #1:

The authors have provided sufficiently detailed answers to the points raised. I recommend it for publishing.

Referee #3:

Overall, the authors have responded well to the first review and the revised version of this manuscript is improved. The additional analyses of the second meiosis I exit dataset generally support the previous conclusions and significantly strengthen the paper.

There is one statement at the top of page 8 that is not supported by the data. The authors report 19% of sites are dephosphorylated during mitotic exit (Fig 2A), and either 14% (replicate 1, Fig 2A) or 21% (replicate 2, Appendix S1B) of sites are dephosphorylated during meiosis I exit. These numbers suggest that roughly the same amount of dephosphorylation occurs during exit from mitosis and meiosis I, but the authors conclude "dephosphorylation seems to be predominant only during mitotic, and not meiotic, exit". This statement should be changed or removed.

All editorial and formatting issues were resolved by the authors.

Dr. Sandra A Touati
CNRS - Institut Jacques Monod
15 rue Helene Brion
Paris 75013
France

5th Jan 2024

Re: EMBOJ-2023-115446R1

Qualitative rather than quantitative phosphoregulation shapes the end of meiosis I in budding yeast

Dear Dr. Touati,

Thank you for submitting your final revised manuscript for our consideration. I am pleased to inform you that we have now accepted it for publication in The EMBO Journal.

Yours sincerely,

Hartmut Vodermaier
